# Lipid Oxidation at the Crossroads: Oxidative Stress and Neurodegeneration Explored in *Caenorhabditis elegans*

**DOI:** 10.3390/antiox14010078

**Published:** 2025-01-10

**Authors:** Julia Tortajada-Pérez, Andrea del Valle Carranza, Cristina Trujillo-del Río, Mar Collado-Pérez, José María Millán, Gema García-García, Rafael Pascual Vázquez-Manrique

**Affiliations:** 1Laboratory of Molecular, Cellular and Genomic Biomedicine, Instituto de Investigación Sanitaria La Fe, 46026 Valencia, Spain; julia_tortajada@iislafe.es (J.T.-P.); cristina_trujillo@iislafe.es (C.T.-d.R.); mar_collado@iislafe.es (M.C.-P.); jose_millan@iislafe.es (J.M.M.); gema_garcia@iislafe.es (G.G.-G.); 2Joint Unit for Rare Diseases IIS La Fe—CIPF, 46026 Valencia, Spain; 3Center for Biomedical Network Research on Rare Diseases (CIBERER), Instituto de Salud Carlos III, 28029 Madrid, Spain

**Keywords:** lipid metabolism, lipid oxidation, protein misfolding, neurodegenerative disorders, *C. elegans*

## Abstract

Lipid metabolism plays a critical role in maintaining cellular integrity, especially within the nervous system, where lipids support neuronal structure, function, and synaptic plasticity. However, this essential metabolic pathway is highly susceptible to oxidative stress, which can lead to lipid peroxidation, a damaging process induced by reactive oxygen species. Lipid peroxidation generates by-products that disrupt many cellular functions, with a strong impact on proteostasis. In this review, we explore the role of lipid oxidation in protein folding and its associated pathological implications, with a particular focus on findings in neurodegeneration from *Caenorhabditis elegans* studies, an animal model that remains underutilized. Additionally, we highlight the effectiveness of different methodologies applied in this nematode to deepen our understanding of this intricate process. In the nervous system of any animal, including mammals and invertebrates, lipid oxidation can disturb the delicate balance of cellular homeostasis, leading to oxidative stress, the build-up of toxic by-products, and protein misfolding, key factors in neurodegenerative diseases. This disruption contributes to the pathogenesis of neurodegenerative disorders such as Alzheimer’s, Parkinson’s, or Huntington’s disease. The findings from *Caenorhabditis elegans* studies offer valuable insights into these complex processes and highlight potential avenues for developing targeted therapies to mitigate neurodegenerative disease progression.

## 1. Introduction

Neurodegenerative disorders (NDDs) are a complex area in healthcare, characterized by progressive neuronal dysfunction and cell death, which profoundly impact patients’ health and quality of life (reviewed by [1,2,3,4,5]). A central feature of many NDDs is the aggregation of various misfolded proteins, disrupting cellular homeostasis and contributing to neurodegeneration [6]. In this review, we provide a comprehensive overview of how lipid oxidation affects the disrupted protein homeostasis observed in NDDs and discuss the power of *Caenorhabditis elegans* (*C. elegans*) as a model organism for future studies of the underlying mechanisms.

A wide range of NDDs are linked to protein aggregation events, including well-known disorders such as Alzheimer’s (AD) and Parkinson’s diseases (PD), as well as rare conditions like Huntington’s disease (HD) or other cytosine–adenine–guanine (CAG) repeat disorders. Some pathogenic variants can cause proteins to adopt aberrant structures or become prone to aggregation, leading to disruptions in essential cellular processes and triggering stress responses that may result in irreversible cellular damage or cell death [6]. Thus, understanding the processes contributing to protein aggregation is crucial for elucidating the disease pathogenesis and developing therapeutic strategies that can be applied to a wide range of diseases with this common feature.

In this regard, lipid oxidation is a compelling and often overlooked area of focus that calls for deeper examination. Exacerbated lipid oxidation has been linked to enhanced cellular proteotoxicity and accelerated tissue degeneration. Lipids play key roles in the central nervous system, which extend structural support within cellular membranes, acting as crucial facilitators of neuronal signal transduction processes, ensuring efficient communication between neurons by modulating their fluidity and thermodynamics and acting as signaling molecules or their precursors [7]. However, when lipid oxidation is unbalanced, these functions may be compromised, impacting overall neuronal health.

*C. elegans* has been widely used as a model organism for studying NDDs for several reasons, including the fact that conducting similar experiments in mammalian systems is substantially more complex and presents greater ethical challenges. A simple nervous system, such as that of *C. elegans*, comprising a well-defined set of neurons [8,9,10], enables detailed analysis of neuronal function and pathology. This nematode also serves as a robust platform for screening potential therapeutic compounds targeting key enzymes, including metabolism regulators such as AMPK and LET-363/TOR, sirtuins, and nuclear receptors [11,12,13,14,15], among many other examples. These enzymes coordinately regulate cellular health by engaging in processes such as energy homeostasis, autophagy, stress resistance, and lifespan extension, making them central to interventions aimed at improving metabolic function and mitigating age-related pathologies. More importantly, *C. elegans* provides powerful tools for direct and indirect genetic analysis, allowing researchers to dissect pathways and identify key modulators of cellular processes. It is particularly suitable for studying molecular processes and interactions in vivo. These experiments are significantly more time-consuming and costly, if not outright impossible, in vertebrates.

In this review, we aim to explore the dual role of lipid oxidation in neurotoxicity within these diseases, both by exacerbating protein misfolding and damaging lipids, which disrupt their structural integrity and impair their functions. We further emphasize the utility of *C. elegans* models as a straightforward yet powerful platform for investigating this phenomenon while also addressing the inherent limitations associated with this model organism. Understanding the intricate relationship between lipid oxidation and neuronal function offers insights into specifically targeting the oxidation of lipids for alleviating neuronal dysfunction and slow disease progression, holding promise for improving outcomes in patients with NDD.

### 1.1. Brief Overview of Neurodegenerative Diseases Caused by Protein Aggregation

Proteins must adopt specific structures to perform their designated functions, and achieving these structures depends on the tightly regulated process of protein folding, which is largely determined by the protein’s sequence [6]. When proteins fail to fold correctly, a cascade of detrimental effects is triggered within cells, as these misfolded proteins may lose their normal biological functions, interrupting essential cellular processes. Furthermore, misfolded proteins can acquire toxic properties and form insoluble aggregates, dragging other cellular components with them or both, disrupting cellular pathways and contributing to cellular damage and death. As numerous studies have shown, protein misfolding, oligomerization, and accumulation in the brain are the main events triggering pathology in these diseases [6,16,17]. Furthermore, the burden exerted by misfolded proteins triggers cellular stress responses, which, if persistent, can exacerbate dysfunction and accelerate disease progression.

NDDs characterized by protein aggregation can manifest in both familial and sporadic forms. Familial cases are often linked to specific genetic mutations that predispose individuals to protein aggregation, while sporadic forms arise from a combination of genetic susceptibility, environmental factors, and age-related changes in protein homeostasis. In the hereditary forms of this disease, mutations in genes, including single-nucleotide polymorphisms, alternative splicing variants, and repeat expansions, can result in the production of proteins that are prone to aggregation, which in turn enhances their collapse within aggregates. Notable examples are HD, caused by an expansion in the huntingtin gene (*HTT*), and certain familial forms of amyotrophic lateral sclerosis (ALS) that are considered monogenic, associated with mutations in *SOD1* (superoxide dismutase), *TARDBP* (Transactive response DNA-binding protein), or *FUS* [18]. Similar genetic mutations are also implicated in some spinocerebellar ataxias. However, AD and PD, as well as other forms of ALS, among others, have more complex and multifactorial origins involving both genetic factors and non-genetic influences, such as changes in their environment, translational errors, and dysregulation of protein folding systems and machinery. Therefore, modeling them for a study can become a challenge. To overcome this issue, neurodegeneration caused by protein aggregation can be achieved by taking advantage of the molecular causes of monogenic diseases. For example, models of HD are very easy to create by simply expressing the DNA encoding mutant huntingtin (mHtt), or even a small fragment of it (see below), that contains the prone-to-aggregate toxic peptide. Additionally, the genetic forms of AD and PD can also be modeled ectopically, introducing the mutant proteins in different animal models, from worms [19,20,21,22,23] to rodents [24,25,26].

HD is a hereditary autosomal dominant neurodegenerative disorder caused by CAG repeat expansion in the first exon of the *HTT* gene, encoding the huntingtin protein (Htt) [27]. This expansion encodes a polyglutamine (polyQ) track, which makes mHtt prone to aggregation. Moreover, through caspase cleavage and alternative splicing, N-terminal fragments, which are more toxic than full-length mHtt, also collapse into intracellular inclusions [28,29,30]. Additionally, the peptide encoded by the first exon, including the abnormally long CAG expansion, is believed to be the most toxic form of mHtt. HD patients experience progressive motor dysfunction, cognitive decline, and psychiatric symptoms due to the degeneration of striatal medium spiny neurons, typically occurring over 15–20 years [31,32,33].

In contrast, PD and AD are more prevalent but also more complex NDDs with multifactorial causes that range from genetic predispositions to environmental influences. PD is characterized by tremor, rigidity, and bradykinesia, with postural instability typically emerging in the advanced stages of the disease [34]. It features the intracellular accumulation of Lewy bodies and neurites composed of aggregated α-synuclein, parkin, and ubiquitinated proteins [35]. α-synuclein, which is mostly unfolded in the brain, adopts a β-sheet-rich amyloid-like structure prone to aggregation [36,37]. The disease primarily affects nigrostriatal dopaminergic neurons, with age being the greatest risk factor [38]. In a similar manner, AD is defined by the progressive loss of memory and cognitive functions due to the accumulation of β-amyloid (Aβ) plaques and neurofibrillary tau tangles. Tau, a protein essential for stabilizing microtubules, becomes hyperphosphorylated and forms aggregates [39]. AD is the most common type of dementia, accounting for two-thirds of dementia cases in those over 65.

Each one of these diseases is characterized by the misfolding of a specific protein: mHtt in HD, Aβ and tau in AD, and α-synuclein in PD, which aggregates into insoluble inclusions within neurons [40]. Protein aggregates disrupt cellular homeostasis because there is a loss of the natural function of aggregation-prone molecules, but also because the aggregates sequester other essential proteins, interfering with many other cellular processes, which ultimately leads to neuronal death [41,42], as mentioned in Hipp and Hartl [43]. At the same time, impaired mitochondrial function, also shared by HD [44], AD [45], and PD [46], leads to reduced ATP production, depriving neurons of the energy necessary for their survival (see Clemente-Suárez et al. for a comprehensive review [47]). Excessive reactive oxygen species (ROS) production contributes to oxidative stress, causing damage to many cellular molecules and components, including lipids, proteins, and DNA, while promoting further neuronal dysfunction and cell death (reviewed by Chen et al. [48]). Altogether, protein aggregation, oxidative stress, and the dysfunction of relevant cellular processes lead to the widespread disruption of proteostasis.

In response to the accumulation of misfolded proteins and other cellular stressors, cells typically activate these depurating pathways and other stress response pathways; however, in HD, AD, and PD, these protective mechanisms become dysregulated, both in patients (see [49,50,51]) and in mammalian models [52,53]. These events cause the chronic activation of stress responses, namely persistent endoplasmic reticulum stress, and can lead to apoptosis, as cells are unable to cope with the ongoing damage.

The shared biomolecular and cellular features of HD, AD, and PD highlight the central role of protein aggregation in driving neurodegeneration, regardless of the specific protein involved. Understanding these common mechanisms is crucial for identifying potential therapeutic targets to prevent or slow neurodegeneration in these diverse diseases.

### 1.2. Comprehensive View of the Protein Homeostasis Network in NDDs

To prevent and regulate protein aggregation, cells rely on an extensive proteostasis network that includes molecular chaperones and mechanisms for degradation and clearance. The two primary clearance pathways are the ubiquitin–proteasome system and the autophagy–lysosome pathway (hereafter referred to as autophagy). Both of them play critical roles in maintaining protein homeostasis, as detailed in Wilson et al. [54]. The ubiquitin–proteasome system primarily degrades ubiquitin-marked short-lived proteins [52,55,56,57]. In contrast, the autophagy pathway is responsible for clearing unfolded long-lived proteins, protein aggregates, and defective organelles, including damaged mitochondria through mitophagy [58], by enclosing cellular material within the autophagosome, a double-lipidic membrane structure [54]. Nevertheless, there is strong cross-talk between both pathways, given that ubiquitin labeling also occurs in autophagy to degrade specific targets [59]. Notably, the autophagy pathway also acts at the axons [60] and synapses [61] of neurons, where it contributes to local protein homeostasis and mitophagy [58], which is required for effective synaptic function.

These mechanisms are activated under stress conditions, such as low energy, limited amino acid availability, or the collapse of misfolded proteins, enabling the clearance of the damaged proteins and organelles (reviewed by Singh and Cuervo [62]). Therefore, prone-to-aggregation proteins should naturally be candidates for degradation via the autophagy pathway. However, in NDDs, these cellular mechanisms responsible for maintaining healthy proteins become substantially compromised [56,63]. Dysregulated autophagy and lysosomal function are tightly linked to cell death pathways, leading to neuronal death when proteostasis is heavily disrupted [54]. This impairment, in turn, not only prevents the removal of other damaged components elsewhere but also increases the accumulation of toxic protein aggregates, creating a self-perpetuating feedback loop.

In addition to the ubiquitin–proteasome system and autophagy pathways, molecular chaperones and other quality control mechanisms specific to subcellular compartments, such as endoplasmic reticulum-associated degradation and Golgi-associated degradation, also play crucial roles in maintaining protein homeostasis.

Interestingly, beyond their most described functions, lipids and their related pathways, like lipid synthesis and lipid oxidation, also influence proteostasis [11,64], highlighting the complexity of cellular protein regulation. Lipids and proteins are intimately connected at the cellular level, cooperating in numerous processes and forming essential cellular structures. Among these, lipid assemblies, such as lipid rafts (cholesterol-rich membrane microdomains that serve as platforms for signaling complexes) and caveolae (specialized flask-shaped invaginations in the plasma membrane) play pivotal roles in organizing cellular functions. These assemblies play a key role in regulating essential processes like signaling, protein folding, and degradation, all of which are vital for maintaining protein balance. Lipid assemblies affect where proteins are located and how they function, particularly in neurodegenerative pathways. The signaling within these assemblies not only controls protein aggregation but also interacts with mechanisms that regulate aging, such as the insulin/IGF-1 pathway [64,65,66]. Beyond this, lipid droplets (LDs) also play a significant role in managing proteotoxic stress, particularly when the proteasomal system is overwhelmed by interacting with deposition sites for protein aggregates and facilitating their clearance [67]. In terms of metabolic pathways, lipid synthesis, for instance, contributes to the formation of membrane structures necessary for autophagy and endosomal trafficking, which are critical for protein clearance. On the other hand, lipid peroxidation can modify proteins, promoting aggregation or impairing their degradation. Additionally, certain lipid types, such as sphingolipids and cholesterol, influence membrane dynamics and protein interactions, directly affecting the efficiency of proteostasis pathways.

All these cellular mechanisms orchestrate the intricate balance of protein synthesis, folding, and degradation, collectively maintaining proteostasis and preserving cellular health. Alterations in lipid composition, often observed in NDDs, can disrupt the delicate balance required for processes like autophagosome formation or lysosomal function. These findings suggest that lipid metabolism and lipid-related oxidative processes are deeply interconnected with protein homeostasis and that understanding the interplay of these mechanisms is crucial for deciphering the molecular basis of these diseases and developing therapeutic strategies to restore proteostasis and cellular health. In this context, animal models are essential for shedding light on these processes.

### 1.3. The Interplay Between Lipid Oxidation, Proteostasis, and Neuronal Death in NDDs

As mentioned above, oxidative stress is a common feature in pathogenic processes, particularly in NDDs. When oxidative stress occurs, ROS attack critical cellular components, such as proteins, lipids, and nucleic acids, which undergo oxidation, rendering them dysfunctional or as pro-oxidants themselves, establishing an amplification cycle that precipitates cellular death, neuronal loss and, consequently, further oxidative stress. Among these molecules, oxidized lipids not only contribute to the degeneration of neuronal structures but also enhance aggregate accumulation and activate signaling pathways that promote inflammation and cell death.

Oxidation of lipid molecules occurs in two notable biochemical processes: lipid peroxidation and β-oxidation, both of which play critical roles in cellular metabolism, homeostasis, and survival. β-oxidation is an essential and natural catabolic oxidative process that occurs in the mitochondria and involves the breakdown of fatty acids to generate acetyl-CoA, which enters the citric acid cycle for energy production. Instead, lipid peroxidation occurs when oxidants interact with fatty acids, especially polyunsaturated fatty acids (PUFAs) within lipids in an unregulated manner. This reaction is initiated by ROS attacking the double bonds, followed by hydrogen detachment from a carbon and oxygen insertion, and results in the formation of lipid peroxyl radicals and hydroperoxides [68]. These hydroperoxides can further react with other lipids, proteins, or DNA, resulting in the generation of reactive intermediates such as lipid peroxyl radicals and reactive aldehydes. This process can also be mediated by enzymes, such as lipoxygenases, cyclooxygenases, and cytochrome P450 [69].

Lipid peroxides can exacerbate protein misfolding by promoting oxidative modifications of proteins, which can alter their structure and function. When proteins misfold, hydrophobic regions of cysteine residues can be exposed on the protein surface, making them vulnerable to oxidation by ROS and other oxidants [70]. Oxidation of cysteine residues can result in the formation of disulfide bonds and mixed disulfide bonds. Additionally, oxidation can disrupt non-covalent interactions within proteins, cause peptide chain fragmentation, promote protein cross-linking, and oxidize specific side chains. These changes collectively contribute to protein destabilization and further misfolding [71]. In turn, these misfolded proteins can further disrupt cellular homeostasis by interfering with processes involved in maintaining proteostasis, such as the ubiquitin–proteasome system and autophagy. On the other hand, lipid peroxidation has long been recognized to disrupt bilayer structures, altering key membrane properties, including fluidity, permeability to various substances, bilayer thickness, and membrane integrity [72,73], and even leading to different types of cell death: (1) apoptosis, (2) death lead by autophagy, and (3) ferroptosis [74,75], as reviewed by [76].

Regarding membrane properties, membrane fluidity stands out as a significant factor for neuropathology since neurons rely on it for proper synaptic function, signal transduction, and vesicle trafficking [77]. Membrane proteins’ structure and function are influenced by the composition, structure, and dynamics of their lipid environment. In alignment with this, it has been widely reported that lipid peroxidation reduces membrane fluidity and disrupts asymmetry by oxidizing membrane components (e.g., oxidized PUFAs and promoting cross-link formation) [78,79].

The products of lipid peroxidation interact with membrane receptors and transcription factors/repressors to induce signaling for apoptosis. For instance, when phosphatidylserine is peroxidized, it is exposed on the external layer, a key feature on apoptotic cell membranes [70,71,72,73,74,75,76,77,78,79,80,81,82,83], as the loss of lipid asymmetry [84]. On the other hand, apoptotic cells are an additional source of oxidized phospholipids and may actively contribute to inflammation [85].

Moreover, lipid peroxidation leads to ferroptosis, a form of iron-dependent cell death characterized by the accumulation of lethal lipid peroxides. Ferroptosis begins with the accumulation of ferrous iron (Fe^2+^) within cells, which can be derived from several sources, including cellular uptake via transferrin receptors and the release from ferritin stores. The excess of iron is critical because it catalyses the formation of ROS through the Fenton reaction, where iron reacts with hydrogen peroxide to produce highly reactive hydroxyl radicals (Figure 1). The initial oxidative stress induces the peroxidation of PUFAs present in cellular membranes and generates lipid hydroperoxides, which can further decompose into a variety of reactive aldehydes, such as 4-hydroxynonenal. As lipid peroxidation progresses, reactive lipid species, e.g., lipid-ROS, accumulate within the cell, a hallmark of ferroptosis (Figure 1). Ultimately, this process leads to the disruption of membrane integrity and function, causing the cell to die [86,87].

The interplay between lipid oxidation, protein misfolding, and cellular death is particularly relevant in neurodegenerative diseases, where both processes are commonly observed. For instance, in conditions such as HD, AD, and PD, the accumulation of misfolded proteins, i.e., mHtt, Aβ, tau, and α-synuclein, respectively, is associated with increased oxidative stress and mitochondrial dysfunction. Lipid peroxidation products can further exacerbate protein aggregation, while misfolded proteins can impair lipid metabolism, creating a toxic environment that drives the progression of these diseases. Regarding apoptosis, ROS generated during β-oxidation can damage mitochondrial membranes, leading to the release of cytochrome C and the subsequent activation of the apoptotic cascade. Meanwhile, the accumulation of lipid peroxides during ferroptosis underscores the lethal potential of disrupted lipid homeostasis. Thus, understanding these processes offers insight into how alterations in lipid metabolism can drive cell death, contributing to the pathophysiology of NDD. This connection highlights the convergence of lipid oxidation and protein homeostasis in the regulation of cell survival and death.

As previously discussed, maintaining proteostasis throughout life is a key cellular strategy against neurodegeneration. However, while much is known about the roles of protein chaperones and other mechanisms in proteostasis, the contributions of lipids remain less explored. Lipid assemblies like lipid rafts and caveolae are not only structural entities but also active participants in regulating proteostasis, aging pathways, and neurodegenerative mechanisms. Furthermore, LDs contribute to the clearance of protein aggregates and regulation of cellular homeostasis. Thus, dysregulation in lipid metabolism, particularly changes in the oxidative states of lipids, can disrupt these processes, damaging the proteostasis network and accelerating neurodegenerative disease progression.

### 1.4. Key Genes Linking Lipid Dysregulation to NDDs

Certain genes associated with lipid metabolism, particularly lipid oxidation, have been implicated as risk factors or contributors to neurodegeneration. Variants of these genes, as well as isoforms of the proteins they encode, have been identified in patients and experimental models exhibiting neurodegeneration alongside disturbances in lipid metabolism, including elevated lipid oxidation and peroxidation, as highlighted below.

For example, a well-known variant of the apolipoprotein E gene (*APOE*), apoE4, has been described to pose a strong risk factor for AD [88,89,90]. The APOE protein is implicated in lipid and fatty acid metabolism, and this variant has been associated with an impairment of these processes [91]. Notably, apoE4 has also been shown to increase nitric oxide release in humanized APOE mice and human microglia [92], while metabolic shifts towards lipid oxidation have been observed in humanized mice expressing apoE4 [93]. Another example is the *PARK7* gene encoding the protein DJ-1. DJ-1 has been associated with the development of PD and participates in oxidative stress protection [94,95]. Likewise, a mutation in the *SCP2* gene, encoding the SCPx enzyme, was present in a patient who exhibited progressive neurodegeneration, cardiac dysrhythmia, and metabolic abnormalities, including altered fatty acid levels and disrupted β-oxidation pathways. Pharmacological treatments like fenofibrate and 4-hydroxytamoxifen increased SCPx levels and improved certain metabolic markers, suggesting potential therapeutic strategies for SCPx deficiency [96]. Similarly, *LRRK2*, which encodes the leucine-rich repeat kinase 2, has also been associated with PD. *LRRK2* is involved in lipid metabolism as it regulates the carnitine palmitoyltransferase 1A (CPT1A), the critical enzyme of β-oxidation.

Concerning enzymes involved in the peroxidation process, changes in their sequence or expression levels have been associated with neurodegeneration. Take, for example, the cyclooxygenase 2 gene (*COX2*), an enzyme of the peroxidation pathway, whose enhanced expression has been associated with several neurodegenerative diseases, such as PD [97,98,99], AD [100,101], and ALS [102,103]. Similarly, the mammalian reticulocyte 15-LOX-1 is the major enzyme responsible for membrane lipid peroxidation and metabolite apoptosis inducers [104]. This enzyme is widely expressed in the CNS [105], and increased activity of 15-LOX-1 has been observed in aged brains during inflammation and neurodegenerative diseases such as AD [106,107]. These studies highlight the significant role of genes involved in lipid metabolism, particularly those regulating lipid oxidation and peroxidation, in the development of neurodegenerative diseases. Variants in genes such as *APOE*, *PARK7*, *SCP2*, and *LRRK2*, along with alterations in key enzymes like COX and 15-LOX-1, demonstrate how disruptions in lipid metabolic processes contribute to neurodegeneration. These insights highlight the intricate relationship between lipid metabolism and neurodegenerative diseases, offering valuable directions for future research and potential therapeutic strategies aimed at modulating lipid oxidation and related pathways.

## 2. The Role of Lipid Peroxidation in Neurodegeneration: Insights from *C. elegans* Models

### 2.1. Advantages and Challenges of Using C. elegans for Lipid Peroxidation Studies

Many characteristics render the nematode *C. elegans* a valuable model organism for exploring the interplay between lipid peroxidation and the neuronal dysfunction seen in NDDs. Its simplicity, genetic tractability, and highly conserved biochemical pathways establish it as an excellent system for such studies. Notably, its well-characterized nervous system, composed of 302 neurons, including a precisely defined dopaminergic network of eight neurons [8], provides a robust platform for modelling specific processes and assessing their impact on neuronal function. Additionally, its short life cycle, compared to other models, enables rapid observation of phenotype progression as well as the generation of consecutive generations within a few weeks (Figure 2A).

Several *C. elegans* models recapitulate the key pathological features of human NDDs (Figure 2B), allowing for detailed mechanistic studies. In AD, transgenic *C. elegans* expressing human Aβ peptides have been developed, which aggregate and cause progressive neurotoxicity [19]. These models exhibit behavioral deficits, neuronal dysfunction, and oxidative damage, presenting somehow similar traits to the pathological hallmarks of AD. Similarly, models expressing human α-synuclein in dopaminergic neurons mimic the dopaminergic neuron degeneration observed in PD [108]. In the study of HD, *C. elegans* has become a widely used model to replicate the pathogenic features of polyQ-containing proteins seen in patients. Numerous transgenic *C. elegans* models have been developed to express polyQ proteins in various configurations, including the full-length Htt protein, the first exon of *HTT* [109,110], or isolated polyQ tracts [111,112]. These models exhibit hallmark features of HD, such as protein aggregation in neurons and other cell types, leading to neurodegeneration and behavioral impairments (reduced motility, enabling quantitative assessment of motor deficits that mirror the movement impairments seen in HD patients). The diversity among these models, based on the specific Htt fragment expressed, the length of the polyQ track, or the promoter driving expression, provides distinct tools to study various aspects of HD pathology. Moreover, in other *C. elegans* HD models, the expression of mutant Htt is restricted to specific neuronal populations, such as dopaminergic neurons, allowing for a targeted investigation of how different neuron types respond to polyQ toxicity. This tissue-specific approach facilitates a more nuanced understanding of the differential vulnerability of neurons to mHtt-induced damage, particularly in neuronal circuits involved in motor control or sensory processing.

Beyond its applications in modeling neurodegenerative diseases, *C. elegans* is also a valuable tool for studying lipid metabolism and peroxidation processes, owing to its genetic tractability and well-characterized pathways. Mutants deficient in enzymes involved in PUFA biosynthesis provide valuable insights into how disruptions in lipid metabolism contribute to increased lipid peroxidation and oxidative stress. Additionally, these mutants help elucidate the role of specific lipids in cellular protection and damage by impairing or facilitating their production. *C. elegans* is also a valuable model for lipidomic research, considering that many of its lipids metabolic pathways are conserved with higher organisms, including humans. Its lipidome has been extensively studied and has been comprehensively characterized. Moreover, there are antibodies specifically developed for use in *C. elegans* research; techniques, such as immunofluorescence, Western blotting, immunoprecipitation, ELISA, liquid chromatography, high-resolution mass spectrometry (2D-LC/HRMS), and flow cytometry, are also available for worms. However, there are limitations to consider when using *C. elegans* as a model. Isolating specific cells or tissues for biochemical analyses is challenging, though not impossible, as techniques such as single-worm [113], single-tissue [114], and single-cell approaches [115] have been developed. Moreover, *C. elegans* lacks some tissues and organs found in vertebrates, such as the eyes, heart (although the pharynx is considered an analogous structure), and liver, which is functionally substituted by its intestine. Some neurodegeneration models in *C. elegans* are also considered artefactual, as the toxic gene products being studied do not naturally exist in these organisms. In addition, gene redundancy poses challenges; some mammal and human genes have many orthologs in *C. elegans* with non-identical functions, making it harder to generate a complete lack of function of the products of the orthologue genes. While the homology between *C. elegans* and higher organisms is high, it is not identical, which can limit direct comparisons in certain contexts.

### 2.2. Current Understanding of Lipid Peroxidation in Neurodegenerative Diseases from C. elegans Models

As has been discussed previously in this review (see Section 1.3), lipid oxidation has a critical impact on neurodegeneration. Lipids, as fundamental components of cellular membranes and signaling pathways, are crucial for maintaining central nervous system health. However, oxidative stress can severely disrupt their normal qualities by driving excessive lipid peroxidation, a process that compromises membrane integrity, alters signaling cascades, and exacerbates protein misfolding events. Dysregulated lipid–protein interactions further contribute to these misfolding processes, promoting the fibrillogenic and amyloidogenic processing of disease-specific protein isoforms, such as Aβ in AD, Htt in HD, and α-synuclein in PD [116]. Consequently, enhancing lipid metabolism can thus improve the efficacy of these systems, facilitating the removal of damaged or misfolded proteins [117].

The role of glial LDs in this context has been explored more recently. Mutations in key lipolysis genes in *C. elegans* have been shown to lead to the appearance of LDs in neurons, which offer protection from hyperactivation-triggered neurodegeneration. This protection is associated with a mild reduction in touch sensation. Additionally, reduced biosynthesis of PUFAs and impaired lipolysis work synergistically to enhance PUFA partitioning into triacylglycerol rather than phospholipids, providing further neuronal protection [118]. Similarly, in *Drosophila*, glial LDs have been shown to protect neuroblasts from oxidative stress, particularly under hypoxic conditions. The incorporation of PUFAs into neutral lipids stored in LDs helps reduce ROS damage by preventing PUFA peroxidation. In response to ROS insult or mitochondrial dysfunction, *Drosophila* neurons increase lipid production via SREBP-mediated lipogenesis, with lipids being transferred to neighboring glia to form LDs [118,119]. Furthermore, in cultured hippocampal neurons, excess fatty acids are transferred to astrocytes via apoE-associated lipid particles, where they are stored in astrocyte LDs and β-oxidized in mitochondria. This β-oxidation helps protect neurons from oxidative stress and maintain energy balance during periods of enhanced neuronal activity [120]. These findings suggest that glial LDs play a protective role for neurons by reducing oxidative stress, although it remains unclear why neurons do not autonomously form LDs under stress conditions. Moreover, in several mutant *C. elegans* strains, it has been reported that lipid peroxidation induces changes in PUFA content [121], thus promoting membrane rigidity.

Modulating the lipid state by targeting lipid oxidation has been shown to effectively prevent protein misfolding-associated toxicity in numerous *C. elegans* models of neurodegeneration (Table 1). For instance, in PD, neuroprotection has been linked to the mobilization of fatty acids and triglycerides stored in LDs, which help mitigate oxidative stress. LDs can mitigate oxidative stress induced by 6-hydroxydopamine (6-OHDA), suggesting that controlled lipid accumulation under stress conditions may offer neuroprotection. This process has been shown to reduce damage and improve cellular function by restoring lipid homeostasis [118]. Similarly, cannabidiol has demonstrated beneficial effects in PD models by reducing lipid peroxidation, promoting lipid deposition, and enhancing proteostasis through the upregulation of the *sod-3/SOD2* (superoxide dismutase 3) gene, which mitigates oxidative stress and improves neuronal survival [122]. In HD, antioxidant interventions such as *Epimedium polysaccharide* (*EbPS-A1*) have alleviated lipid peroxidation and reduced ROS production, leading to improved survival in polyQ nematode models exposed to oxidative stress from paraquat [123]. These findings underscore the importance of managing lipid oxidation to reduce neurodegeneration in polyQ-induced toxicity. Additionally, in AD, benzofuran derivatives have shown promise by reducing lipid peroxidation and preventing Aβ aggregation, which is a hallmark of the disease. These compounds also help to restore acetylcholine levels, thereby preventing cholinergic neuronal degeneration and reducing oxidative stress [124]. A more novel mechanism has been identified in ferroptosis, a form of cell death triggered by lipid peroxidation. Studies have shown that dihomo-γ-linolenic acid, a PUFA, induces neurodegeneration upon conversion to dihydroxyeicosadienoic acid, highlighting a new avenue for understanding lipid-driven neurodegeneration and the role of lipid peroxidation in these diseases [125].

The use of ROS scavenger compounds aimed at alleviating ferroptosis has also been studied in *C. elegans*. Liproxstatin-1 (Lip-1), a potent lipid autoxidation inhibitor which has ROS scavenger properties, has demonstrated significant efficacy in mitigating ferroptosis-related damage in neurodegenerative disease models [126]. Lip-1 effectively inhibits lipid peroxidation downstream of glutathione depletion, a critical process exacerbated during aging that increases susceptibility to ferroptotic cell death. Studies indicate that Lip-1 treatment significantly reduces lipid peroxidation markers, such as malondialdehyde and 4-hydroxynonenal, while attenuating age-related ferroptotic cell death in the intestinal cells of the worms. Additionally, Lip-1 extends organismal lifespan by approximately 70% and alleviates late-life frailty, highlighting its potential to improve healthspan by targeting ferroptosis, a mechanism implicated in neurodegeneration [127]. Also, pharmacological intervention with ferrostatin-1, another lipid peroxidation inhibitor, mitigated increased mortality, elevated lipid peroxidation, and reduced GPX4 (glutathione peroxidase 4) activity and morphological damage to *C. elegans*’ dopaminergic neurons, showing therapeutic potential [128]. Furthermore, using the UA44 worm strain, which overexpresses alpha-synuclein in cherry-labeled dopaminergic neurons, Fe^2+^ administration caused similar alterations in wild-type animals, linking ferroptosis and dopamine signaling in a Parkinsonian phenotype. These findings underscore the potential of targeting ferroptosis to alleviate the physiological, biochemical, and morphological consequences of Fe^2+^ overload while encouraging further exploration of genetic and dopamine-mediated effects in PD contexts [128]. Likewise, Schlotterer et al. (2021) examined the protective effects of sulforaphane, an indirect antioxidant, and vitamin E (alpha-tocopherol), a direct antioxidant, against glucotoxicity [129]. Hyperglycemia-induced conditions led to increased ROS and methylglyoxal-derived advanced glycation end products, causing neuronal damage and reduced lifespan. Treatment with sulforaphane (20 µmol/L) and Vitamin E (200 µg/mL) prevented ROS increase, advanced glycation end product accumulation, and preserved neuronal function, maintaining lifespan similar to controls. These results suggest that both sulforaphane and vitamin E may offer therapeutic potential for mitigating glucotoxicity and preventing neurodegeneration.

Dietary fatty acids have been proposed in the regulation of ferroptosis. Dihomo-γ-linolenic acid (DGLA) has been identified as a potent inducer of ferroptosis in the germ cells of *C. elegans*. In this model, co-treatment with ferrostatin-1 significantly alleviates both germ cell death and sterility, underscoring the role of ferroptosis in these processes, too [130]. Additionally, the incorporation of monounsaturated fatty acids (MUFAs), such as oleic acid, either exogenously or through genetic manipulation, provides protection by displacing PUFAs like DGLA from cellular membranes, thereby reducing lipid ROS accumulation. In a study that identified ferroptosis as a key driver of iron-overload-induced damage in both cultured cells and *C. elegans* models, oleic acid protected against this damage potentially through lipidomic reprogramming involving the modulation of phospholipid composition [131]. The protective effects of oleic acid were linked to nuclear hormone receptors, such as NHR-49/PPAR-α.

Beyond lipid oxidation, interventions aimed at enhancing lipid synthesis have also been explored for their neuroprotective effects. In PD, *H. leucospilota extracts* and *HLEA -P1 compounds* have been shown to restore lipid deposition, reduce protein aggregation, and reduce oxidative stress, often through transcriptional regulation pathways involving DAF-16/FOXO and HSF-1/HSF-1 [132,133]. Additionally, mitochondrial lipid metabolism, particularly β-oxidation, has emerged as a key player in neuroprotection. For instance, in HD and ALS, enhancing mitochondrial β-oxidation via *acdh-1/ACADM* (Acyl CoA Dehydrogenase) and *kat-1/ACAA1* (3-keto fatty acyl-CoAs) upregulation restores lipid homeostasis and energy balance, thereby mitigating neurodegeneration [134]. Furthermore, novel pathways, such as ferroptosis, a form of cell death triggered by lipid peroxidation products, have been identified as contributors to neurodegeneration in *C. elegans* models, offering promising new avenues for therapeutic exploration [131]. These findings underscore the critical relationship between lipid oxidation, lipid synthesis, and neurodegeneration.

**Table 1 antioxidants-14-00078-t001:** Reported antioxidant studies on NDDs focusing on restoring lipid deposits/metabolism and their protective effects of proteostasis.

Metabolic Processes	Pathology	Key Findings	Intervention	Strain Generation	Applied Technologies	References
Oxidative stress	Alzheimer’s Disease	Decrease in lipid deposition and reduction in Aβ aggregation and oxidative stress. Rise in ACh levels (preventing cholinergic neuronal degeneration).	Benzofuran derivatives and chalcones	Conventional genetics	Microscopy	[124]
Oxidative stress	Alzheimer’s disease and polyQs-induced toxicity	Recalibration of lipid metabolism with an increase in the expression of genes involved in fatty acid β-oxidation, restoration of innate immune, and detoxification responses. Activation of SKN-1 leading to stress resistance. Increase in life span and improvement of age-related physical fitness together with the rescue of HD- and AD-like behavioral deficits.	*AbaPep#07* (derived from abalone hemocyanin)	Conventional genetics	Microscopy, RNA-seq, RT-PCR, and GC–MS	[135]
Oxidative stress	Polyglutamine-induced neurotoxicity	Alleviation of lipid peroxidation and ROS production. Increases in the survival rates of polyQ nematodes intoxicated by paraquat by boosting antioxidant defenses.	Epimedium polysaccharide (EbPS-A1)	Conventional genetics	Microscopy and spectrophotometry	[123]
Oxidative stress	Parkinson’s disease	Restoration of lipid content in transgenic worms expressing α-synuclein. Recovery of DAergic neurons after 6-OHDA-induced neurodegeneration and a significant decrease in α-synuclein aggregation. Reduction in intracellular ROS through the activation of DAF-16 transcription factor (*sod-3*, *hsp16.1*, *hsp16.2*, and *hsp12.6*).	HLEA-P1 compound- Decanoic acid	Conventional genetics	Microscopy, spectrophotometry, and RT-qPCR	[132]
Oxidative stress	Parkinson’s disease	Reduction in malondialdehyde content (inhibition of lipid peroxidation) and increase in SOD and GPx activities. Mitigation of oxidative stress by regulating apoptosis and restoring the function of the cholinergic system.	*Astragalan*	Conventional genetics	Microscopy, spectrophotometry, and RT-PCR	[136]
Oxidative stress	Parkinson’s disease	Alleviation of lipid level changes, α-synuclein aggregation, improvement of locomotory behavior, and augmentation of dopamine levels. Increase in mRNA expression of *daf-16*, *sod-1*, *sod-3*, and *ctl-2* and downregulation of *sod-2*, resulting in lifespan extension.	*Tambulin*	Conventional genetics	Microscopy, spectrophotometry, LC-MS/MS, and qPCR	[137]
Oxidative stress	Parkinson’s disease	Restoration of lipid deposition through upregulation of *fat-7* and enhanced *gcs-1*-mediated glutathione synthesis. Potential activation mediated by HSF-1 and DAF-16 transcription factors.	2-butoxy tetrahydrofuran (2-BTHF)	Conventional genetics, transgenesis	Molecular biology, in silico molecular coupling, RNA Seq, RT-qPCR, and microscopy	[138]
Oxidative stress	Parkinson’s disease	RAC1/*ced-10* mutants displayed an increased unsaturation index, suggesting an increase in the content of polyunsaturated fatty acid (PUFA) and lipo-oxidative damage. *ced-10* mutation produces a decrease in dopaminergic function and an elevation in the number of autophagic vesicles.	RAC1/*ced-10* gene	Conventional genetics	Microscopy, synchrotron radiation, µFourier-transform infrared spectroscopy (SR-µFTIR), and UHPLC-MS	[117]
Oxidative stress	Parkinson’s disease	Significant decrease in fat content and reduction in alpha-synuclein aggregation. Positive regulation of *sod-1*, *sod-2*, *sod-3*, *gst-4*, *gst-7*, and *ctl-2*. Decrease in protein carbonyl content.	*Shatavarin* IV	Conventional genetics	Microscopy, RT-PCR, and LC-MS/MS	[139]
Oxidative stress	Parkinson’s disease	Increase in lipid accumulation as lipid droplets and protection against 6-hydroxydopamine (6-OHDA)-induced degeneration.	High-glucose and high-fructose diets	Transgenesis via extrachromosomal arrays, fluorescent protein tagging, and conventional genetics	Microscopy and qPCR	[140]
Oxidative stress	Parkinson’s disease	Recovery of lipid deposition and reduction in α-synuclein aggregation. Neuroprotection, food-sensing improvement, and lifespan extension. Upregulation of *cat-2* (DA-synthesis) and *sod-3* (free-radical scavenging) and downregulation of *egl-1* (apoptosis).	*H. leucospilota* body wall and cuveirian tubule extracts	Conventional genetics	Microscopy, RT-PCR, and H-NMR	[133]
Oxidative stress	Neurodegeneration mediated by glucotoxicity	The combination of compounds prevented the rise in ROS and the accumulation of methylglyoxal-derived advanced glycation end products and safeguarded neuronal function, maintaining lifespan at levels similar to the wild-type strain.	Sulforaphane (SFN) and vitamin E (alpha-tocopherol)	Conventional genetics	Microscopy and LC/MS-MS	[129]
Lipid peroxidation	Parkinson’s disease	Reduction in lipid peroxidation and the increment of lipid depositions trigger the induction of the ubiquitin-like proteasome and autophagy flux and reduce oxidative stress by upregulation of *sod-3* expression.	Cannabidiol	Transgenesis and conventional genetics	Microscopy and immunofluorescence	[122]
Lipid peroxidation	Parkinson’s disease	Neuroprotective effect due to the mobilization of fatty acids and triglycerides in excess.	Lipid droplets	Conventional genetics and transgenesis CRISPR	Molecular biology, single nucleotide polymorphism (SNP) mapping, microscopy, and HPLC	[118]
Ferroptosis	Parkinson’s disease and ferroptosis-mediated neurodegeneration	Ferrostatin-1 elevated lipid peroxidation, mitigated increased mortality, and reduced GPX4 activity and morphological damage to dopaminergic neurons.	Ferrostatin-1	Conventional genetics	Microscopy and spectrophotometry	[128]
Ferroptosis	Ferroptosis-mediated neurodegeneration	DGLA triggers neurodegeneration upon conversion to dihydroxyeicosadienoic acid through the action of CYP-EH (CYP, cytochrome P450; EH, epoxide hydrolase), representing a new class of lipid metabolites that induce neurodegeneration via ferroptosis.	Dihomo-γ-linolenic acid (DGLA)	Transgenesis, fluorescent protein tagging, and conventional genetics	Microscopy and LC/MS-MS	[125]
Ferroptosis	Ferroptosis-mediated neurodegeneration	Ferrostatin-1 provides protection by displacing polyunsaturated fatty acids (PUFAs) from cellular membranes, thus decreasing the buildup of lipid-derived ROS. Furthermore, it notably reduces germ cell death and sterility caused by DGLA. Additionally, oleic acid demonstrates protective effects, which are linked to the nuclear hormone receptor NHR-49/PPAR-α.	Ferrostatin-1, Oleic acid	Conventional genetics and transgenesis	Fluorescence microscopy	[130,131]
Ferroptosis	Ferroptosis-mediated neurodegeneration	Lip-1 significantly lowers lipid peroxidation markers, such as malondialdehyde (MDA) and 4-hydroxynonenal (4-HNE), reducing age-related ferroptotic cell death in intestinal cells and enhancing healthspan.	Liproxstatin-1 (Lip-1)	Conventional genetics	Spectrophotometry, microscopy, X-ray fluorescence microscopy, and LC-inductively coupled plasma MS	[127]
β-oxidation	ALS and Huntington’s disease	Restoration of lipid homeostasis, lipid accumulation, and energy balance through mitochondrial β-oxidation by upregulation of *acdh-1* and *kat-1* (fatty acid metabolism and β-oxidation).	*L. rhamnosus* HA-114	Conventional genetics and transgenesis	Microscopy and RNA-Seq analysis (LC-MS)	[134]

## 3. Potential Interventions and Future Perspectives

Currently, there are two main strategies for therapy development: one that boosts the antioxidant capacity of the experimental model and, therefore, impacts metabolism and lipid deposits, and another that focuses on the possibility of lipid-based therapies in combatting neurodegenerative diseases.

Antioxidant strategies to reduce oxidative stress have long been used to treat NDDs (see Forman and Zhang for a review [141]. As stated above, oxidative stress is a key driver of neuronal damage and is essential in mitigating the progression of NDDs (see Houldsworth for a review [142]). These antioxidant approaches include the use of antioxidant compounds, gene therapy, and dietary interventions aimed at increasing the activity of endogenous antioxidant enzymes. The regulation of peroxidation pathways can achieve the restoration of lipid balance. These interventions not only combat oxidative damage but also address the metabolic imbalances that contribute to cellular vulnerability in neurodegenerative conditions. Among the antioxidant options available, there are numerous compounds that use different strategies. Notable examples include chemical chaperones like 4-phenylbutyrate [143], activators of the cAMP-PKA pathway such as *tambulin* [137], natural extracts such as *shatavarin* [139], and phenolic compounds such as cannabidiols [122].

On the other hand, therapies based on lipid supplementation imply the use of lipids, which are usually dysregulated in neurodegenerative pathologies, acting as signaling molecules that regulate cell survival, apoptosis (programmed cell death), and inflammatory responses. One current area of focus is the use of omega-3 fatty acids, which have demonstrated neuroprotective effects. These PUFAs, found in fish oil and certain plant oils, have been shown to reduce inflammation, promote neuronal survival, and enhance synaptic plasticity. Clinical studies suggest that omega-3 supplementation may slow cognitive decline in AD patients and improve motor function in individuals with PD [144]. Another promising approach involves the modulation of sphingolipid metabolism. Sphingolipids are crucial for cell signaling and membrane integrity, as described by Alessenko et al. [145]. The report indicates that certain sphingolipid metabolites can promote neuroprotection and reduce apoptosis in neuronal cells. Therapies aimed at restoring sphingolipids balance may provide a novel avenue for treating neurodegeneration.

By targeting both antioxidant capacity and lipid metabolic stability, these strategies offer a multifaceted approach to potentially slow or even reverse neurodegenerative processes.

## 4. Discussion

Lipid peroxidation has emerged as a critical factor in the pathogenesis of neurodegenerative disorders, serving as a convergence point for oxidative stress and cellular damage. Processes such as ferroptosis, a regulated form of cell death driven by iron-dependent lipid peroxidation, highlight the vulnerability of neurons to oxidative insults [68]. In the context of NDDs, the accumulation of lipid peroxides contributes to the disruption of membrane integrity, reducing fluidity and impairing cellular signaling, both of which are essential for neuronal function and survival [78]. Additionally, it promotes protein misfolding [146], which can lead to a feedback loop that exacerbates cellular damage.

Mutations in genes associated with lipid metabolism, antioxidant defenses, and iron homeostasis have been linked to heightened oxidative stress and accelerated neuronal damage. For instance, the enhanced expression of enzymes involved in lipid peroxidation pathways, such as cyclooxygenases, has been identified in models of neurodegenerative diseases [98,99]. Moreover, the use of lipids as biomarkers for NDD progression has already been suggested [147]; nonetheless, the heterogeneity of NDDs necessitates a personalized approach to therapeutic development, as the contribution of lipid peroxidation may vary across different diseases and patient populations.

These findings suggest that dysregulated lipid peroxidation may not only be a hallmark of neuronal damage but also act as a driving force for disease progression. Intervention strategies targeting lipid peroxidation have shown promise in mitigating neurodegenerative processes. ROS-scavenger compounds, including ferrostatins, have demonstrated efficacy in reducing ferroptosis and preserving neuronal integrity [128]. These compounds function by neutralizing lipid radicals and preventing the propagation of peroxidative damage, offering a potential therapeutic avenue for halting or slowing disease progression. Nevertheless, challenges remain in translating these findings into clinical applications. The complex interplay between lipid peroxidation and other pathological processes, such as protein aggregation and mitochondrial dysfunction, warrants further investigation.

In this regard, *C. elegans* offers several advantages as a model organism compared to other animal models to study a range of phenotypes and mechanisms in neurodegenerative research. *C. elegans* is a very small model and can be grown in large numbers on Petri dishes, making it significantly easier and less expensive to maintain than vertebrates [148,149]. With a short 3-day life cycle and a lifespan of about three weeks, these animals offer a significant advantage for research, as most other organisms have longer lifespans, making experiments more time-consuming and costly (Figure 2A). Moreover, *C. elegans* show really high fertility since a single worm can produce about 250–300 offspring. *C. elegans* allow for the quick generation of results and multi-generational studies. The large number of *C. elegans* makes this model suitable for high-throughput drug screening, which is more challenging with mice.

To date, several genetic alterations (deletions, point mutations, inversions, etc.) are available in this nematode, which are very useful to study gene function, disease mechanisms, and other biological processes. Several NDDs can be modeled by introducing gene knockouts, point mutations, and/or small insertions/deletions using CRISPR Cas-9 for genome editing. Also, using RNA interference (RNAi) to silence specific gene expression temporarily is helpful in producing models for human diseases. One advantage of RNAi is that it generally reduces but does not completely eliminate gene expression when the complete knockout is lethal. For example, Friedreich’s ataxia, which happens in humans when frataxin expression is reduced but not completely ablated (because it results in lethality), can be modeled using the knockdown of the *fhr-1* gene [150,151], the worm orthologue of human *FXN*, the gene that encodes frataxin. RNAi can be induced by transgenes that produce a hairpin or simply by feeding with a strain of *E. coli* that expresses dsRNA of the target gene [152]. Gene overexpression is also a good resource and is typically done by introducing extra copies of a gene into *C. elegans* using transgenic constructs. In *C. elegans*, if the exogenous DNA is not inserted within a chromosome, an event which is rare becomes arranged in the so-called extrachromosomal arrays [153]. These arrays act as satellite chromosomes that are inherited in a non-Mendelian way. Overexpression can also be done using promoters that drive the expression in a tissue-specific manner. Reporter gene fusions are also possible to visualize where and when specific genes are active, providing spatial and temporal gene expression data. Figure 2B displays several examples of neurodegenerative models created in *C. elegans*. As already exemplified in Table 1, there are multiple tools to apply to the model in the study of neurodegenerative diseases, making it a versatile and powerful model for genetic and biomedical research. Various genetic manipulation techniques can be employed to generate new model strains in addition to those already available in the Caenorhabditis Genetics Center (CGC) repository.

This formidable toolkit, which makes the worm useful for modelling neurodegeneration, can also be exploited to investigate oxidative stress. For example, removing genes that are active in pathways that produce free radical scavengers, like the biosynthetic pathways that regulate glutathione [154], or genes that catalyse neutralization of these radicals, such as superoxide dismutase, etc. [155]. Additionally, using screens and systems of mass investigation of many variables and situations, like using multiwalled plates [156], automated microscopes [157], or worm flow equipment [158]. Furthermore, of course, all types and kinds of the so-called OMIC approaches can be used to study oxidative stress, lipid oxidation, and neurodegeneration in these small and not-so-humble animals [11,147,159,160]. Similarly, various quantification methodologies are readily applicable to this model, including fluorescence microscopy, electron microscopy, nuclear magnetic resonance, and liquid chromatography or gas chromatography. These last techniques are particularly important for the identification and quantification of oxidative biomarkers.

## 5. Conclusions

Lipid peroxidation plays a central role in the pathogenesis of NDDs, acting as a key driver of oxidative stress and cellular damage. Its contributions to processes like ferroptosis, protein misfolding, and disrupted neuronal signaling underscore its critical importance as both a marker and a therapeutic target.

As demonstrated by this review, the ease of generating mutant *C. elegans* strains for nearly any gene of interest establishes this model as a powerful platform for investigating phenotypes and their underlying pathways. Furthermore, the capacity to integrate these genetic models with pharmacological treatments and dietary supplementation assays further enhances the versatility of *C. elegans* as a research tool. This combined approach facilitates the systematic exploration of gene–environment interactions and the evaluation of potential therapeutic interventions, providing valuable insights into disease mechanisms and treatment strategies.

By leveraging this model, the field can make significant progress in understanding the interplay between lipid peroxidation and neurodegeneration. Future research should focus on integrating advanced methodologies, such as OMICs approaches and high-resolution imaging performed in patients, to further elucidate the role of lipid peroxidation and its downstream effects. These efforts have the potential to identify novel therapeutic targets and develop personalized interventions that mitigate lipid peroxidation, ultimately slowing the progression of neurodegenerative disorders.

## Figures and Tables

**Figure 1 antioxidants-14-00078-f001:**
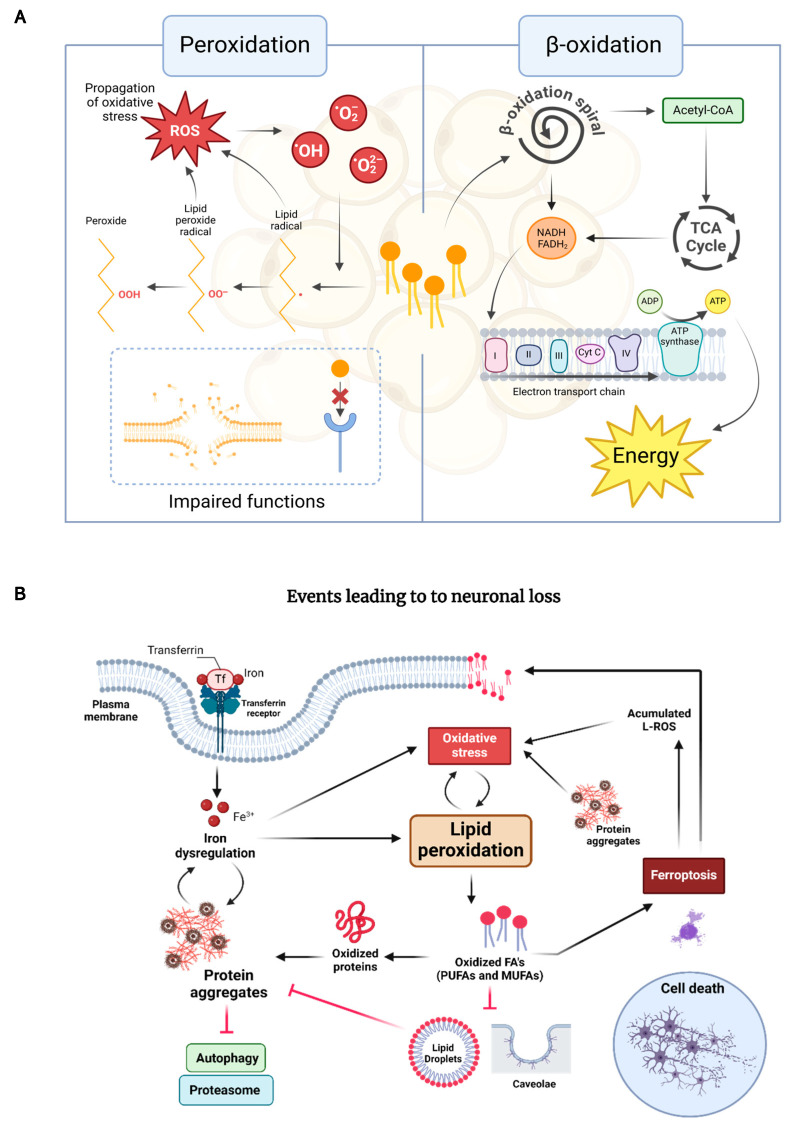
Processes that contribute to lipid oxidation within cells drive neuronal death. (**A**) Lipid peroxidation and β-oxidation are distinct processes involving lipid oxidation, each with different outcomes. In peroxidation, ROS drives oxidation, resulting in the formation of various oxidized species that impair cellular functions. In contrast, β-oxidation involves the breakdown of lipids into smaller molecules through the β-oxidation spiral and the tricarboxylic acid cycle (TCA) cycle, generating energy for the cell. (**B**) Iron enters cells via transferrin–receptor endocytosis, releasing Fe^3+^, which is reduced to Fe^2+^. In the cytoplasm, Fe^2+^ undergoes the Fenton reaction with hydrogen peroxide (H_2_O_2_), generating highly reactive hydroxyl radicals (•OH). These radicals initiate lipid peroxidation, damaging PUFAs in cellular membranes and producing toxic by-products. This oxidative stress disrupts membranes, impairs autophagic flow and mitochondrial function, and damages proteins and DNA, triggering neuronal death through apoptosis and inflammation, a process implicated in a range of neurodegenerative diseases.

**Figure 2 antioxidants-14-00078-f002:**
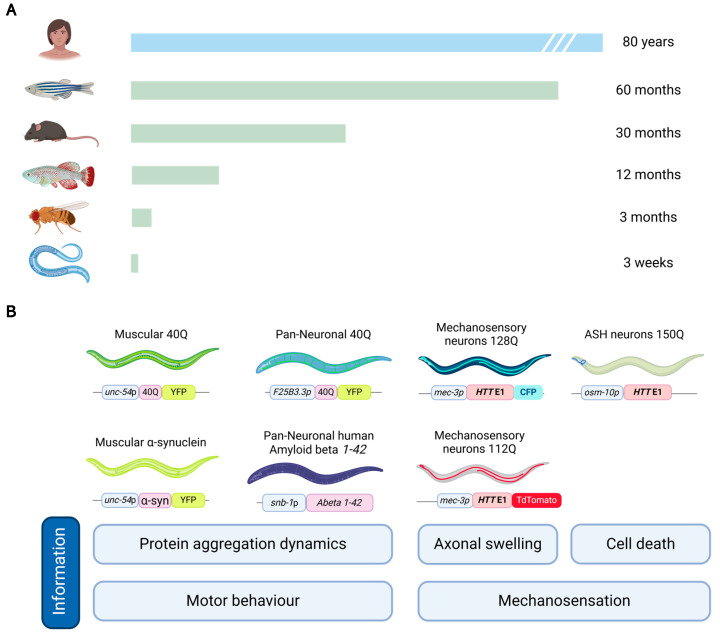
*C. elegans* is a very useful animal model used to research neurodegeneration. (**A**) The nematode *C. elegans* has one of the shortest lifespans among commonly used animal models in basic and applied research. With a lifespan of approximately three weeks, it offers a significant advantage over models such as *Drosophila* (three months) and various vertebrates, whose lifespans range from one to several years. (**B**) Several neurodegenerative disease models have been established in *C. elegans*, including polyQ- and α-synuclein-induced toxicity models, where aggregation-prone proteins are fused to fluorescent tags to visualize in vivo aggregation dynamics. Other models express panneuronally distributed polyQ repeats or β-amyloid, which induce motor behavior impairments and allow for monitoring of protein aggregation. Additionally, models expressing mHtt in mechanosensory neurons are used to assess neuronal functionality, while the expression of mutant proteins in ASH neurons, which are bimodal (responding to both mechanical and chemical stimuli), leads to cell death.

## Data Availability

No data are available for sharing in this manuscript.

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
