# Peer review of "Lipid Oxidation at the Crossroads: Oxidative Stress and Neurodegeneration Explored in *Caenorhabditis elegans"

_antioxidants, 2025, doi:10.3390/antiox14010078_

Round 1
Reviewer 1 Report
In this review article, Tertajada-Pérez and others discuss the role of lipid oxidation in neurodegenerative diseases. They provide a summary of the literature that covers the role protein aggregation has in neurodegenerative diseases and the impact lipid peroxidation could have in this process. They also provide a summary of the strengths of using the invertebrate model system C. elegans for understanding the pathology that may underly neurodegenerative diseases. Although many reviews have been written that cover protein aggregation in neurodegenerative diseases and C. elegans as a model system, this review aims to cover the less reviewed topic of the impact lipid peroxidation has in protein aggregation and neurodegenerative diseases. The review lays out some of the key findings in Table 1 and gives a brief description of some of these findings in the text. While the authors are trying to highlight the strengths of some of the findings in C. elegans, the most impressive studies discussed come from Drosophila research (e.g., lipid droplets and glia). Nevertheless, it is a comprehensive review; however, it that could be improved. Please see the following suggestions:
One of the critical impacts that lipid peroxidation has on cellular function is on membrane fluidity. This topic is only briefly mentioned and discussion of how this may influence neurodegeneration is lacking.
The authors are attempting to highlight the strengths of C. elegans for the investigation of lipid peroxidation in neurodegeneration. However, it is unclear if any studies have been conducted that utilize lipid ROS scavenger compounds, such as Liproxstain or Ferrostatin or ether lipids in C. elegans neurodegeneration studies.
The authors list strengths of C. elegans but do not mention limitations.
Additionally, many citations are not mentioned.
For example, no refences are cited for the claim on line 216 “Alterations in lipid composition, often observed in NDDs, can delicate balance required for processes like autophagosome formation and lysosomal function.
On line 230-231, it states that “…oxidized lipids not only contribute to the degeneration of neuronal structures but also enhance aggregates accumulation and activate signalling pathways that promote inflammation and cell death” but no evidence or citations are provided.
On line 289-290, no evidence is cited for the following comment: “Lipid assemblies like lipid rafts and caveolae are not only structural entities but also active participants in regulating proteostasis, aging pathways, and neurodegenerative mechanisms.”
On line 334-335, it states that “Its lipidome has been extensively studied and has been comprehensively characterized.” No reference is cited to support this claim.
On line 335-339, no evidence is cited for the following statement: “…there are antibodies specifically developed for use in C. elegans research; techniques as immunofluorescence, western blotting, immunoprecipitation, ELISA, liquid chromatography, high-resolution mass spectrometry (2D-LC/HRMS) and flow cytometry are also available for worms.”
In addition to missing citations, may cited literature in the text is wrong. For example, the references cited on lines 376, 380, 389, 394, 396 are wrong.
Some other textual mistakes are noted:
Line 395, what are acdh-1 and kat-1? What do they encode?
Line 69-70, “This nematode worm serves…”. Nematode worm is redundant.
Line 224, “As previously mentioned on this review…” – mentioned in this review
Line 250, “Lipid peroxidation leads ferroptosis, a form …” - Lipid peroxidation leads to ferroptosis
Line 429, “…NDDs (see Forman and Zhang for a review [71].” - …NDDs (see Forman and Zhang for a review [71]).
Line 429-431 contains an inconsistent sentence.
Line 458-467 contain smaller font then the rest of the text.
Line 463, “(Figure 1A)” should be Figure 2A
Line 505, “All this formidable toolkit…” – should be “This formidable toolkit…”
It is difficult to read Figure 1. Its resolution should be better.
Resolution in Figure 2 could be better. Also, what are the organisms show in Figure 2A?
Author Response
We are gra
We are grateful to the editors and reviewers of our paper for their time and constructive criticisms. We will respond point by point, using blue colour, to highlight our responses.
Referee 1#
Major comments
In this review article, Tortajada-Pérez and others discuss the role of lipid oxidation in neurodegenerative diseases. They provide a summary of the literature that covers the role protein aggregation has in neurodegenerative diseases and the impact lipid peroxidation could have in this process. They also provide a summary of the strengths of using the invertebrate model system C. elegans for understanding the pathology that may underly neurodegenerative diseases. Although many reviews have been written that cover protein aggregation in neurodegenerative diseases and C. elegans as a model system, this review aims to cover the less reviewed topic of the impact lipid peroxidation has in protein aggregation and neurodegenerative diseases.
The review lays out some of the key findings in Table 1 and gives a brief description of some of these findings in the text. While the authors are trying to highlight the strengths of some of the findings in C. elegans, the most impressive studies discussed come from Drosophila research (e.g., lipid droplets and glia). Nevertheless, it is a comprehensive review; however, it that could be improved. Please see the following suggestions:
Lines 351-365: We sincerely thank the referee for their valuable feedback. Our intention with this review was to highlight C. elegans as a valuable tool for exploring the mechanisms underlying lipid peroxidation and neurodegeneration, especially for future studies on the field. However, upon reviewing the manuscript, we realized that including Drosophila studies in the context of LDs in neurons may have caused some confusion, as the focus of our review is on the findings in C. elegans. To address this, we have restructured the section to first present the findings from C. elegans regarding lipid droplet formation and its protective effects in neurons, followed by a brief mention of similar findings in Drosophila and other models like cultured hippocampal neurons. We initially included the studies carried on other models as they were relevant to the broader discussion on lipid droplet dynamics, but we now see that separating these findings more clearly in the context of different models will help avoid confusion. We hope these revisions better clarify our focus on C. elegans while still acknowledging the valuable contributions from other models.
Previous text:
The origin and role of glial LDs have been explored more recently. In Drosophila, glial LDs act as a niche for neuroblasts and protect against oxidative stress, particularly under hypoxic conditions. The incorporation of polyunsaturated fatty acids (PUFAs) into neutral lipids stored in LDs helps reduce ROS damage by preventing the toxic peroxidation of these fatty acids. This process mitigates oxidative stress in neuroblasts, which would otherwise be exacerbated by PUFA peroxidation [66,67]. Additionally, when Drosophila neurons face ROS insult or mitochondrial dysfunction, they increase lipid production through SREBP-mediated lipogenesis. However, instead of forming LDs, lipids are transferred to neighbouring glia to form LDs. In cultured hippocampal neurons, excess fatty acids are transferred to astrocytes via ApoE-associated lipid particles and stored in astrocyte LDs, where they are β-oxidized in mitochondria. This β-oxidation helps protect neurons during periods of enhanced activity by reducing oxidative stress and maintaining cellular energy balance [68]. These findings suggest that glial LDs play a protective role for neurons by reducing oxidative stress, although it remains unclear why neurons do not autonomously form LDs under stress conditions.
Revised version:
The role of glial LDs in this context has been explored more recently. Mutations in key lipolysis genes in C. elegans have been shown to lead to the appearance of lipid droplets (LDs) in neurons, which offer protection from hyperactivation-triggered neurodegeneration. This protection is associated with a mild reduction in touch sensation. Additionally, reduced biosynthesis of polyunsaturated fatty acids (PUFAs) and impaired lipolysis work synergistically to enhance PUFA partitioning into triacylglycerol rather than phospholipids, providing further neuronal protection. These findings highlight the critical role of neuronal lipolysis in regulating neuronal function and protecting against neurodegeneration [doi: 10.15252/embr.202050214]
Similarly, in Drosophila, glial LDs have been shown to protect neuroblasts from oxidative stress, particularly under hypoxic conditions. The incorporation of PUFAs into neutral lipids stored in LDs helps reduce ROS damage by preventing PUFA peroxidation. In response to ROS insult or mitochondrial dysfunction, Drosophila neurons increase lipid production via SREBP-mediated lipogenesis, with lipids being transferred to neighbouring glia to form LDs [66,67]. Furthermore, in cultured hippocampal neurons, excess fatty acids are transferred to astrocytes via ApoE--associated lipid particles, where they are stored in astrocyte LDs and β-oxidized in mitochondria. This β-oxidation helps protect neurons from oxidative stress and maintain energy balance during periods of enhanced neuronal activity[68]. These findings suggest that glial LDs play a protective role for neurons by reducing oxidative stress, although it remains unclear why neurons do not autonomously form LDs under stress conditions
One of the critical impacts that lipid peroxidation has on cellular function is on membrane fluidity. This topic is only briefly mentioned and discussion of how this may influence neurodegeneration is lacking.
We appreciate the referee’s insightful suggestion regarding the critical impact of lipid peroxidation on membrane fluidity and its influence on neurodegeneration. While we acknowledge the importance of this topic, our review was designed to focus more extensively on the interplay between lipid peroxidation, protein misfolding and ferroptosis. As such, we decided not to expand on the membrane fluidity aspect in greater detail. That being said, we fully agree that the link between lipid peroxidation-induced changes in membrane fluidity and neurodegeneration is a valuable topic that would certainly enrich the review. To acknowledge it, we have briefly expanded the discussion in the revised manuscript to highlight how changes in membrane fluidity can contribute to neurodegenerative processes.
Lines 262-284:
On the other hand, lipid peroxidation has long been recognized to disrupt bilayer structures, altering key membrane properties, including fluidity, permeability to various substances, bilayer thickness, and membrane integrity [DOI: 10.1016/0005-2736(81)90284-4; DOI: 10.1016/0891-5849(88)90011-1]; and even leading to different types of cell death —(1) apoptosis , (2) death lead by autophagy and (3) ferroptosis (https://doi.org/10.1016/j.taap.2017.11.006, https://doi.org/10.1021/bi100517x, and reviewed by https://doi.org/10.3389/fcell.2023.1226044).
Regarding membrane properties, membrane fluidity stands out as significant factor for neuropathology, since neurons rely on it for proper synaptic function, signal transduction, and vesicle trafficking (REF). Membrane proteins’ structure and function is influenced by the composition, structure and dynamics of its lipid environment. In alignment with this, it has been widely reported that lipid peroxidation reduces membrane fluidity and disrupts asymmetry by oxidizing membrane components (e.g., oxidized PUFAs) and promoting cross-link formation [DOI: 10.1016/0891-5849(94)90167-8, DOI: 10.1007/978-1-4020-8831-5_13]. .
The products of lipid peroxidation interact with membrane receptors and transcription factors/repressors to induce signalling for apoptosis. For instance, when phosphatidylserine is peroxidized, it is exposed on the external layer, a key feature on apoptotic cell membranes (Borisenko et al., 2004; Fadok et al., 1992; Matsura et al., 2005; Tyurina et al., 2004b), as the loss of lipid asymmetry (Savill and Fadok, 2000). On the other hand, apoptotic cells are an additional source of oxidized phospholipids and may actively contribute to inflammation (Huber et al., 2002).
+ Lines 434-438:
“Moreover, in several mutant C. elegans strains, it has been reported that lipid peroxidation induces changes in PUFAs content (https://doi.org/10.1093/genetics/iyab093), thus, also promoting membrane rigidity.”
The authors are attempting to highlight the strengths of C. elegans for the investigation of lipid peroxidation in neurodegeneration. However, it is unclear if any studies have been conducted that utilize lipid ROS scavenger compounds, such as Liproxstain or Ferrostatin or ether lipids in C. elegans neurodegeneration studies.
We thank the referee for this insightful comment. In response, we have incorporated additional references and information into the manuscript regarding studies that utilize Liproxstatin-1 and Ferrostatin-1 and other compounds as ROS scavengers in different C. elegans models. Specifically, we highlight their role in mitigating lipid peroxidation and protecting against neuronal dysfunction. This update aims to provide a more comprehensive discussion of the pathogenic role of peroxidation, and ferroptosis, in neurodegeneration and how using these compounds has therapeutic potential. Moreover, we want to emphasize the usefulness of C. elegans in these purposes.
Revised text: Lines 466- 509:
“The use of ROS scavenger compounds aimed at alleviating ferroptosis has also been studied in C. elegans. Liproxstatin-1 (Lip-1), a potent lipid autoxidation inhibitor which has ROS scavenger properties, has demonstrated significant efficacy in mitigating ferroptosis-related damage in neurodegenerative disease models [https://doi.org/10.1016/j.intimp.2022.108770. Lip-1 effectively inhibits lipid peroxidation downstream of glutathione depletion, a critical process exacerbated during aging that increases susceptibility to ferroptotic cell death. Studies indicate that Lip-1 treatment significantly reduces lipid peroxidation markers, such as malondialdehyde (MDA) and 4-hydroxynonenal (4-HNE), while attenuating age-related ferroptotic cell death in intestinal cells of the worms. Additionally, Lip-1 extends organismal lifespan by approximately 70% and alleviates late-life frailty, highlighting its potential to improve healthspan by targeting ferroptosis, a mechanism implicated in neurodegeneration (doi: 10.7554/eLife.56580). Also, pharmacological intervention with ferrostatin-1, another lipid peroxidation inhibitor, mitigated inceased mortality, elevated lipid peroxidation, reduced GPX4 activity, and morphological damage to C. elegans’ dopaminergic neurons, showing therapeutic potential https://pubmed.ncbi.nlm.nih.gov/38754108/]. Furthermore, using the UA44 worm strain, which overexpresses alpha-synuclein in cherry-labeled dopaminergic neurons, Fe²⁺ administration caused similar alterations in wild-type animals, linking ferroptosis and dopamine signalling in a Parkinsonian phenotype. These findings underscore the potential of targeting ferroptosis to alleviate the physiological, biochemical, and morphological consequences of Fe²⁺ overload, while encouraging further exploration of genetic and dopamine-mediated effects in Parkinsonian contexts[ https://pubmed.ncbi.nlm.nih.gov/38754108/]. Likewise, Schlotterer et al. (2021) examined the protective effects of Sulforaphane (SFN), an indirect antioxidant, and Vitamin E (alpha-tocopherol), a direct antioxidant, against glucotoxicity [doi: 10.1055/a-1158-9248]. Hyperglycemia-induced conditions led to increased ROS and methylglyoxal-derived AGEs, causing neuronal damage and reduced lifespan. Treatment with SFN (20 µmol/l) and Vitamin E (200 µg/ml) prevented ROS increase, AGE accumulation, and preserved neuronal function, maintaining lifespan similar to controls. These results suggest that both SFN and Vitamin E may offer therapeutic potential for mitigating glucotoxicity and preventing neurodegeneration.
Dietary FA have been proposed in the regulation of ferroptosis. Dihomo-γ-linolenic acid (DGLA) has been identified as a potent inducer of ferroptosis in germ cells of C. elegans. In this model, co-treatment with ferrostatin-1, significantly alleviates both germ cell death and sterility, underscoring the role of ferroptosis in these processes too [https://pmc.ncbi.nlm.nih.gov/articles/PMC7483868/]. Additionally, the incorporation of monounsaturated fatty acids (MUFAs), such as oleic acid (OA), either exogenously or through genetic manipulation, provides protection by displacing PUFAs like DGLA from cellular membranes, thereby reducing lipid ROS accumulation. In a study that identifies ferroptosis as a key driver of iron-overload-induced damage in both cultured cells and C. elegans models, OA protects against this damage potentially through a lipidomic reprogramming involving the modulation of phospholipid composition [https://doi.org/10.1016/j.chembiol.2023.10.012]. The protective effects of oleic acid were linked to nuclear hormone receptors, such as nhr-49/PPAR-α. “
The authors list strengths of C. elegans but do not mention limitations.
Lines 348-360: We sincerely appreciate the referee’s valuable suggestions. In response, we have expanded the section discussing the limitations of the model organism, such as the challenges in isolating specific cells or tissues for biochemical analyses, the absence of certain vertebrate tissues and organs, the potential for artifactual neurodegeneration models, and the gene redundancy issue that complicates knockout generation. In addition, we have stated its short life cycle, which enables rapid phenotype progression and the generation of consecutive generations within a few weeks. We believe these additions provide a more balanced and comprehensive discussion of the strengths and limitations of C. elegans in studying neurodegenerative diseases.
Text added:
However, there are limitations to consider when using C. elegans as a model. Isolating specific cells or tissues for biochemical analyses is challenging, though not impossible, as techniques such as single-worm and single-cell approaches have been developed. Moreover, C. elegans lacks some tissues and organs found in vertebrates, such as eyes and a heart—though the pharynx is considered an analogous structure—and a liver, which is functionally substituted by its intestine. Some neurodegeneration models in C. elegans are also considered artefactual, as the toxic gene products being studied do not naturally exist in these organisms. In addition, gene redundancy poses challenges; some mammal and human genes have many orthologs in C. elegans with non-identical functions, making it harder to generate knockouts. While the homology between C. elegans and higher organisms is high, it is not identical, which can limit direct comparisons in certain contexts."
Detail comments
Additionally, many citations are not mentioned.
For example, no refences are cited for the claim on line 216 “Alterations in lipid composition, often observed in NDDs, can delicate balance required for processes like autophagosome formation and lysosomal function.
The referee is right, and hence we have included the following references at the end of the text:
doi: 10.1083/jcb.200505082
doi: 10.1038/s41598-023-44203-6
10.3390/ijms25021125
On line 230-231, it states that “…oxidized lipids not only contribute to the degeneration of neuronal structures but also enhance aggregates accumulation and activate signalling pathways that promote inflammation and cell death” but no evidence or citations are provided.
We agree with the referee and we have cited the following literature, as examples of the message of this text:
PMID: 34064409
DOI: 10.1016/j.redox.2012.10.003
DOI: 10.1016/j.chemphyslip.2006.02.006
On line 289-290, no evidence is cited for the following comment: “Lipid assemblies like lipid rafts and caveolae are not only structural entities but also active participants in regulating proteostasis, aging pathways, and neurodegenerative mechanisms.”
We have followed the suggestion of the referee citing the following literature:
PMID: 37526691
PMID: 20585024
DOI: 10.1016/j.devcel.2023.02.004
On line 334-335, it states that “Its lipidome has been extensively studied and has been comprehensively characterized.” No reference is cited to support this claim.
Same as above we citate these original articles:
PMID: 36320604
PMID: 33594638
And two reviews of C. elegans lipidomics
DOI: 10.1016/j.trac.2023.117374
DOI: 10.1016/j.abb.2015.06.003
On line 335-339, no evidence is cited for the following statement: “…there are antibodies specifically developed for use in C. elegans research; techniques as immunofluorescence, western blotting, immunoprecipitation, ELISA, liquid chromatography, high-resolution mass spectrometry (2D-LC/HRMS) and flow cytometry are also available for worms.”
We have rephrased the text as follows:
“…there are antibodies specifically developed for use in C. elegans research (PMID: 20405020); techniques as immunofluorescence (DOI: 10.1073/pnas.79.5.1558), western blotting (DOI: 10.1016/0014-5793(91)80335-Z), immunoprecipitation (PMID: 32510481), ELISA (DOI: 10.3390/molecules27123858), liquid chromatography (DOI: 10.3390/molecules28145373), high-resolution mass spectrometry (2D-LC/HRMS) (DOI: 10.1002/cpps.114 ; PMID: 36320604 ) and flow cytometry (DOI: 10.1038/nprot.2006.283) are also available for worms.”
In addition to missing citations, may cited literature in the text is wrong. For example, the references cited on lines 376, 380, 389, 394, 396 are wrong.
We apologise for this mistake, and we thank the referee for pointing it out. We have fixed it.
Some other textual mistakes are noted:
Line 395, what are acdh-1 and kat-1? What do they encode?
We have revised the manuscript to include the human orthologs for all C. elegans genes mentioned. Additionally, we have carefully reviewed and standardized the format for both C. elegans and human gene names throughout the manuscript to ensure consistency and compliance with accepted nomenclature guidelines.
We appreciate your attention to detail, and we hope these changes enhance the clarity and readability of the manuscript.
Line 69-70, “This nematode worm serves…”. Nematode worm is redundant.
We have addressed this as suggested by the referee.
Line 224, “As previously mentioned on this review…” – mentioned in this review
We have addressed this as suggested by the referee.
Line 250, “Lipid peroxidation leads ferroptosis, a form …” - Lipid peroxidation leads to ferroptosis
We have addressed this as suggested by the referee.
Line 429, “…NDDs (see Forman and Zhang for a review [71].” - …NDDs (see Forman and Zhang for a review [71]).
We have addressed this as suggested by the referee.
Line 429-431 contains an inconsistent sentence.
We have addressed this as suggested by the referee.
Line 458-467 contain smaller font then the rest of the text.
We have addressed this as suggested by the referee.
Line 463, “(Figure 1A)” should be Figure 2A
We have addressed this as suggested by the referee.
Line 505, “All this formidable toolkit…” – should be “This formidable toolkit…”
We have addressed this as suggested by the referee.
It is difficult to read Figure 1. Its resolution should be better.
Resolution in Figure 2 could be better. Also, what are the organisms show in Figure 2A?
We have improved the resolution of Figure 1 and made some changes for better understanding, as well as added another panel, as suggested by another referee.
teful to the editors and reviewers of our paper for their time and constructive criticisms. We will respond point by point, using blue colour, to highlight our responses.
Referee 1#
Major comments
In this review article, Tortajada-Pérez and others discuss the role of lipid oxidation in neurodegenerative diseases. They provide a summary of the literature that covers the role protein aggregation has in neurodegenerative diseases and the impact lipid peroxidation could have in this process. They also provide a summary of the strengths of using the invertebrate model system C. elegans for understanding the pathology that may underly neurodegenerative diseases. Although many reviews have been written that cover protein aggregation in neurodegenerative diseases and C. elegans as a model system, this review aims to cover the less reviewed topic of the impact lipid peroxidation has in protein aggregation and neurodegenerative diseases.
The review lays out some of the key findings in Table 1 and gives a brief description of some of these findings in the text. While the authors are trying to highlight the strengths of some of the findings in C. elegans, the most impressive studies discussed come from Drosophila research (e.g., lipid droplets and glia). Nevertheless, it is a comprehensive review; however, it that could be improved. Please see the following suggestions:
Lines 351-365: We sincerely thank the referee for their valuable feedback. Our intention with this review was to highlight C. elegans as a valuable tool for exploring the mechanisms underlying lipid peroxidation and neurodegeneration, especially for future studies on the field. However, upon reviewing the manuscript, we realized that including Drosophila studies in the context of LDs in neurons may have caused some confusion, as the focus of our review is on the findings in C. elegans. To address this, we have restructured the section to first present the findings from C. elegans regarding lipid droplet formation and its protective effects in neurons, followed by a brief mention of similar findings in Drosophila and other models like cultured hippocampal neurons. We initially included the studies carried on other models as they were relevant to the broader discussion on lipid droplet dynamics, but we now see that separating these findings more clearly in the context of different models will help avoid confusion. We hope these revisions better clarify our focus on C. elegans while still acknowledging the valuable contributions from other models.
Previous text:
The origin and role of glial LDs have been explored more recently. In Drosophila, glial LDs act as a niche for neuroblasts and protect against oxidative stress, particularly under hypoxic conditions. The incorporation of polyunsaturated fatty acids (PUFAs) into neutral lipids stored in LDs helps reduce ROS damage by preventing the toxic peroxidation of these fatty acids. This process mitigates oxidative stress in neuroblasts, which would otherwise be exacerbated by PUFA peroxidation [66,67]. Additionally, when Drosophila neurons face ROS insult or mitochondrial dysfunction, they increase lipid production through SREBP-mediated lipogenesis. However, instead of forming LDs, lipids are transferred to neighbouring glia to form LDs. In cultured hippocampal neurons, excess fatty acids are transferred to astrocytes via ApoE-associated lipid particles and stored in astrocyte LDs, where they are β-oxidized in mitochondria. This β-oxidation helps protect neurons during periods of enhanced activity by reducing oxidative stress and maintaining cellular energy balance [68]. These findings suggest that glial LDs play a protective role for neurons by reducing oxidative stress, although it remains unclear why neurons do not autonomously form LDs under stress conditions.
Revised version:
The role of glial LDs in this context has been explored more recently. Mutations in key lipolysis genes in C. elegans have been shown to lead to the appearance of lipid droplets (LDs) in neurons, which offer protection from hyperactivation-triggered neurodegeneration. This protection is associated with a mild reduction in touch sensation. Additionally, reduced biosynthesis of polyunsaturated fatty acids (PUFAs) and impaired lipolysis work synergistically to enhance PUFA partitioning into triacylglycerol rather than phospholipids, providing further neuronal protection. These findings highlight the critical role of neuronal lipolysis in regulating neuronal function and protecting against neurodegeneration [doi: 10.15252/embr.202050214]
Similarly, in Drosophila, glial LDs have been shown to protect neuroblasts from oxidative stress, particularly under hypoxic conditions. The incorporation of PUFAs into neutral lipids stored in LDs helps reduce ROS damage by preventing PUFA peroxidation. In response to ROS insult or mitochondrial dysfunction, Drosophila neurons increase lipid production via SREBP-mediated lipogenesis, with lipids being transferred to neighbouring glia to form LDs [66,67]. Furthermore, in cultured hippocampal neurons, excess fatty acids are transferred to astrocytes via ApoE--associated lipid particles, where they are stored in astrocyte LDs and β-oxidized in mitochondria. This β-oxidation helps protect neurons from oxidative stress and maintain energy balance during periods of enhanced neuronal activity[68]. These findings suggest that glial LDs play a protective role for neurons by reducing oxidative stress, although it remains unclear why neurons do not autonomously form LDs under stress conditions
One of the critical impacts that lipid peroxidation has on cellular function is on membrane fluidity. This topic is only briefly mentioned and discussion of how this may influence neurodegeneration is lacking.
We appreciate the referee’s insightful suggestion regarding the critical impact of lipid peroxidation on membrane fluidity and its influence on neurodegeneration. While we acknowledge the importance of this topic, our review was designed to focus more extensively on the interplay between lipid peroxidation, protein misfolding and ferroptosis. As such, we decided not to expand on the membrane fluidity aspect in greater detail. That being said, we fully agree that the link between lipid peroxidation-induced changes in membrane fluidity and neurodegeneration is a valuable topic that would certainly enrich the review. To acknowledge it, we have briefly expanded the discussion in the revised manuscript to highlight how changes in membrane fluidity can contribute to neurodegenerative processes.
Lines 262-284:
On the other hand, lipid peroxidation has long been recognized to disrupt bilayer structures, altering key membrane properties, including fluidity, permeability to various substances, bilayer thickness, and membrane integrity [DOI: 10.1016/0005-2736(81)90284-4; DOI: 10.1016/0891-5849(88)90011-1]; and even leading to different types of cell death —(1) apoptosis , (2) death lead by autophagy and (3) ferroptosis (https://doi.org/10.1016/j.taap.2017.11.006, https://doi.org/10.1021/bi100517x, and reviewed by https://doi.org/10.3389/fcell.2023.1226044).
Regarding membrane properties, membrane fluidity stands out as significant factor for neuropathology, since neurons rely on it for proper synaptic function, signal transduction, and vesicle trafficking (REF). Membrane proteins’ structure and function is influenced by the composition, structure and dynamics of its lipid environment. In alignment with this, it has been widely reported that lipid peroxidation reduces membrane fluidity and disrupts asymmetry by oxidizing membrane components (e.g., oxidized PUFAs) and promoting cross-link formation [DOI: 10.1016/0891-5849(94)90167-8, DOI: 10.1007/978-1-4020-8831-5_13]. .
The products of lipid peroxidation interact with membrane receptors and transcription factors/repressors to induce signalling for apoptosis. For instance, when phosphatidylserine is peroxidized, it is exposed on the external layer, a key feature on apoptotic cell membranes (Borisenko et al., 2004; Fadok et al., 1992; Matsura et al., 2005; Tyurina et al., 2004b), as the loss of lipid asymmetry (Savill and Fadok, 2000). On the other hand, apoptotic cells are an additional source of oxidized phospholipids and may actively contribute to inflammation (Huber et al., 2002).
+ Lines 434-438:
“Moreover, in several mutant C. elegans strains, it has been reported that lipid peroxidation induces changes in PUFAs content (https://doi.org/10.1093/genetics/iyab093), thus, also promoting membrane rigidity.”
The authors are attempting to highlight the strengths of C. elegans for the investigation of lipid peroxidation in neurodegeneration. However, it is unclear if any studies have been conducted that utilize lipid ROS scavenger compounds, such as Liproxstain or Ferrostatin or ether lipids in C. elegans neurodegeneration studies.
We thank the referee for this insightful comment. In response, we have incorporated additional references and information into the manuscript regarding studies that utilize Liproxstatin-1 and Ferrostatin-1 and other compounds as ROS scavengers in different C. elegans models. Specifically, we highlight their role in mitigating lipid peroxidation and protecting against neuronal dysfunction. This update aims to provide a more comprehensive discussion of the pathogenic role of peroxidation, and ferroptosis, in neurodegeneration and how using these compounds has therapeutic potential. Moreover, we want to emphasize the usefulness of C. elegans in these purposes.
Revised text: Lines 466- 509:
“The use of ROS scavenger compounds aimed at alleviating ferroptosis has also been studied in C. elegans. Liproxstatin-1 (Lip-1), a potent lipid autoxidation inhibitor which has ROS scavenger properties, has demonstrated significant efficacy in mitigating ferroptosis-related damage in neurodegenerative disease models [https://doi.org/10.1016/j.intimp.2022.108770. Lip-1 effectively inhibits lipid peroxidation downstream of glutathione depletion, a critical process exacerbated during aging that increases susceptibility to ferroptotic cell death. Studies indicate that Lip-1 treatment significantly reduces lipid peroxidation markers, such as malondialdehyde (MDA) and 4-hydroxynonenal (4-HNE), while attenuating age-related ferroptotic cell death in intestinal cells of the worms. Additionally, Lip-1 extends organismal lifespan by approximately 70% and alleviates late-life frailty, highlighting its potential to improve healthspan by targeting ferroptosis, a mechanism implicated in neurodegeneration (doi: 10.7554/eLife.56580). Also, pharmacological intervention with ferrostatin-1, another lipid peroxidation inhibitor, mitigated inceased mortality, elevated lipid peroxidation, reduced GPX4 activity, and morphological damage to C. elegans’ dopaminergic neurons, showing therapeutic potential https://pubmed.ncbi.nlm.nih.gov/38754108/]. Furthermore, using the UA44 worm strain, which overexpresses alpha-synuclein in cherry-labeled dopaminergic neurons, Fe²⁺ administration caused similar alterations in wild-type animals, linking ferroptosis and dopamine signalling in a Parkinsonian phenotype. These findings underscore the potential of targeting ferroptosis to alleviate the physiological, biochemical, and morphological consequences of Fe²⁺ overload, while encouraging further exploration of genetic and dopamine-mediated effects in Parkinsonian contexts[ https://pubmed.ncbi.nlm.nih.gov/38754108/]. Likewise, Schlotterer et al. (2021) examined the protective effects of Sulforaphane (SFN), an indirect antioxidant, and Vitamin E (alpha-tocopherol), a direct antioxidant, against glucotoxicity [doi: 10.1055/a-1158-9248]. Hyperglycemia-induced conditions led to increased ROS and methylglyoxal-derived AGEs, causing neuronal damage and reduced lifespan. Treatment with SFN (20 µmol/l) and Vitamin E (200 µg/ml) prevented ROS increase, AGE accumulation, and preserved neuronal function, maintaining lifespan similar to controls. These results suggest that both SFN and Vitamin E may offer therapeutic potential for mitigating glucotoxicity and preventing neurodegeneration.
Dietary FA have been proposed in the regulation of ferroptosis. Dihomo-γ-linolenic acid (DGLA) has been identified as a potent inducer of ferroptosis in germ cells of C. elegans. In this model, co-treatment with ferrostatin-1, significantly alleviates both germ cell death and sterility, underscoring the role of ferroptosis in these processes too [https://pmc.ncbi.nlm.nih.gov/articles/PMC7483868/]. Additionally, the incorporation of monounsaturated fatty acids (MUFAs), such as oleic acid (OA), either exogenously or through genetic manipulation, provides protection by displacing PUFAs like DGLA from cellular membranes, thereby reducing lipid ROS accumulation. In a study that identifies ferroptosis as a key driver of iron-overload-induced damage in both cultured cells and C. elegans models, OA protects against this damage potentially through a lipidomic reprogramming involving the modulation of phospholipid composition [https://doi.org/10.1016/j.chembiol.2023.10.012]. The protective effects of oleic acid were linked to nuclear hormone receptors, such as nhr-49/PPAR-α. “
The authors list strengths of C. elegans but do not mention limitations.
Lines 348-360: We sincerely appreciate the referee’s valuable suggestions. In response, we have expanded the section discussing the limitations of the model organism, such as the challenges in isolating specific cells or tissues for biochemical analyses, the absence of certain vertebrate tissues and organs, the potential for artifactual neurodegeneration models, and the gene redundancy issue that complicates knockout generation. In addition, we have stated its short life cycle, which enables rapid phenotype progression and the generation of consecutive generations within a few weeks. We believe these additions provide a more balanced and comprehensive discussion of the strengths and limitations of C. elegans in studying neurodegenerative diseases.
Text added:
However, there are limitations to consider when using C. elegans as a model. Isolating specific cells or tissues for biochemical analyses is challenging, though not impossible, as techniques such as single-worm and single-cell approaches have been developed. Moreover, C. elegans lacks some tissues and organs found in vertebrates, such as eyes and a heart—though the pharynx is considered an analogous structure—and a liver, which is functionally substituted by its intestine. Some neurodegeneration models in C. elegans are also considered artefactual, as the toxic gene products being studied do not naturally exist in these organisms. In addition, gene redundancy poses challenges; some mammal and human genes have many orthologs in C. elegans with non-identical functions, making it harder to generate knockouts. While the homology between C. elegans and higher organisms is high, it is not identical, which can limit direct comparisons in certain contexts."
Detail comments
Additionally, many citations are not mentioned.
For example, no refences are cited for the claim on line 216 “Alterations in lipid composition, often observed in NDDs, can delicate balance required for processes like autophagosome formation and lysosomal function.
The referee is right, and hence we have included the following references at the end of the text:
doi: 10.1083/jcb.200505082
doi: 10.1038/s41598-023-44203-6
10.3390/ijms25021125
On line 230-231, it states that “…oxidized lipids not only contribute to the degeneration of neuronal structures but also enhance aggregates accumulation and activate signalling pathways that promote inflammation and cell death” but no evidence or citations are provided.
We agree with the referee and we have cited the following literature, as examples of the message of this text:
PMID: 34064409
DOI: 10.1016/j.redox.2012.10.003
DOI: 10.1016/j.chemphyslip.2006.02.006
On line 289-290, no evidence is cited for the following comment: “Lipid assemblies like lipid rafts and caveolae are not only structural entities but also active participants in regulating proteostasis, aging pathways, and neurodegenerative mechanisms.”
We have followed the suggestion of the referee citing the following literature:
PMID: 37526691
PMID: 20585024
DOI: 10.1016/j.devcel.2023.02.004
On line 334-335, it states that “Its lipidome has been extensively studied and has been comprehensively characterized.” No reference is cited to support this claim.
Same as above we citate these original articles:
PMID: 36320604
PMID: 33594638
And two reviews of C. elegans lipidomics
DOI: 10.1016/j.trac.2023.117374
DOI: 10.1016/j.abb.2015.06.003
On line 335-339, no evidence is cited for the following statement: “…there are antibodies specifically developed for use in C. elegans research; techniques as immunofluorescence, western blotting, immunoprecipitation, ELISA, liquid chromatography, high-resolution mass spectrometry (2D-LC/HRMS) and flow cytometry are also available for worms.”
We have rephrased the text as follows:
“…there are antibodies specifically developed for use in C. elegans research (PMID: 20405020); techniques as immunofluorescence (DOI: 10.1073/pnas.79.5.1558), western blotting (DOI: 10.1016/0014-5793(91)80335-Z), immunoprecipitation (PMID: 32510481), ELISA (DOI: 10.3390/molecules27123858), liquid chromatography (DOI: 10.3390/molecules28145373), high-resolution mass spectrometry (2D-LC/HRMS) (DOI: 10.1002/cpps.114 ; PMID: 36320604 ) and flow cytometry (DOI: 10.1038/nprot.2006.283) are also available for worms.”
In addition to missing citations, may cited literature in the text is wrong. For example, the references cited on lines 376, 380, 389, 394, 396 are wrong.
We apologise for this mistake, and we thank the referee for pointing it out. We have fixed it.
Some other textual mistakes are noted:
Line 395, what are acdh-1 and kat-1? What do they encode?
We have revised the manuscript to include the human orthologs for all C. elegans genes mentioned. Additionally, we have carefully reviewed and standardized the format for both C. elegans and human gene names throughout the manuscript to ensure consistency and compliance with accepted nomenclature guidelines.
We appreciate your attention to detail, and we hope these changes enhance the clarity and readability of the manuscript.
Line 69-70, “This nematode worm serves…”. Nematode worm is redundant.
We have addressed this as suggested by the referee.
Line 224, “As previously mentioned on this review…” – mentioned in this review
We have addressed this as suggested by the referee.
Line 250, “Lipid peroxidation leads ferroptosis, a form …” - Lipid peroxidation leads to ferroptosis
We have addressed this as suggested by the referee.
Line 429, “…NDDs (see Forman and Zhang for a review [71].” - …NDDs (see Forman and Zhang for a review [71]).
We have addressed this as suggested by the referee.
Line 429-431 contains an inconsistent sentence.
We have addressed this as suggested by the referee.
Line 458-467 contain smaller font then the rest of the text.
We have addressed this as suggested by the referee.
Line 463, “(Figure 1A)” should be Figure 2A
We have addressed this as suggested by the referee.
Line 505, “All this formidable toolkit…” – should be “This formidable toolkit…”
We have addressed this as suggested by the referee.
It is difficult to read Figure 1. Its resolution should be better.
Resolution in Figure 2 could be better. Also, what are the organisms show in Figure 2A?
We have improved the resolution of Figure 1 and made some changes for better understanding, as well as added another panel, as suggested by another referee.

Reviewer 2 Report
This is a timely and informative review of lipid metabolism in C. elegans models of neurodegenerative diseases. It is an important addition to the literature. The focus is mostly on lipid peroxidation, with some info on beta-oxidation. I wonder if having a figure differentiating the two would be informative. Another question is whether lipid oxidation genes themselves are linked to neurodegenerative disorders, directly or as risk factors.
I don't think there is a complete representation of the worm models in Figure 2B, as outlined in Table 1.
Starting at line 351, the section about Drosophila glia seems out of place. Why is Drosophila being discussed here and not worms? If it is because there is limited info about C. elegans glia, then why are flies being discussed and not mammals?
Table 1 should have a title for the first column, and the other titles should have the first letter capitalized.
The text is small for lines 457-467.
Some grammar issues.
line 46-7. Possessives for the disease names are missing.
line 65. The abbreviation for C. elegans should be done earlier.
line 104. Nomenclature for human genes should be consistent (italics for gene, etc.)
line 111. Grammar and syntax issues.
line 278. consistency for beta versus greek symbol for beta
lines 505, 509 Grammar and syntax issues.
Author Response
Referee 2#
Major comments
The usefulness of the model of neurodegenerative diseases using C. elegans has been demonstrated, and although I thought it was a niche, I read it with interest. I think it is well written, but there may be a lack of impartiality in some places because of the overemphasis on the advantages of C. elegans. In addition, I feel that there are many citations of review. For specific phenomena, please cite original research papers as much as possible.
Thank you for your detailed feedback regarding citation in our manuscript. Following your suggestion, we have revised the text to include direct citations wherever possible, drawing upon specific primary sources to support our claims. However, some conclusions in our manuscript represent synthesized insights derived from an extensive number of articles reviewed collectively.
In such cases, citing a comprehensive review is a more realistic and practical approach, as it reflects the broad scope of evidence supporting these conclusions and avoids overloading the manuscript with numerous references. This also provides readers with a single resource to explore the broader context in greater depth.
We appreciate your understanding on this matter and hope that our adjustments strike the appropriate balance between specificity and clarity.
Detail comments
The points that caught my attention are as follows.
The sentences in lines 61 to 62 are redundant
We thank the referee for noticing this mistake, changed it to eliminate redundance .
Previous text: “However, when lipid oxidation is unbalanced, these functions may be compromised, impacting overall neuronal health and signalling be impaired by unbalanced changes in lipid oxidation.”
Revised version: “However, when lipid oxidation is unbalanced, these functions may be compromised, impacting overall neuronal health”
The protein names should be listed their official names, too.
We have corrected this mistake; we thank the referee for noticing it.
I probably understand the arguments in lines 102-107, but since there are also mutations related to aggregate formation in AD and PD, doesn't this sentence create a misunderstanding?
The referee is right. We have rephrased this part to clarify that Alzheimer’s disease (AD) and Parkinson’s disease (PD) have genetic components alongside non-genetic influences, ensuring that the text does not imply a lack of genetic origins for these disorders.
Previous text:
[…] However, other NDDs, like AD and PD among others, have more complex and multifactorial origins [19], including changes in their environment, translational errors, impairment or dysregulation of the protein folding systems and machinery
Revised version:
[…] However, other NDDs, like AD and PD, as well as other sporadic forms of ALS, among others, have more complex and multifactorial origins involving both genetic factors and non-genetic influences such as changes in their environment, translational errors, and dysregulation of protein folding systems and machinery.
Also, it is impossible to deny the possibility that several environmental factors are involved in ALS, etc. Notably, most cases of ALS are sporadic. Many factors may affect the disease onset. I feel that the things the authors want to write about in this part are vague. I recommend that several diseases be discussed separately into sporadic and familial forms.
We meant to allude only at the familiar forms of ALS, as clarified on the manuscript. Here I quote: “This phenomenon is observed in disorders like HD, caused by an expansion in the HTT gene, and certain familial forms of amyotrophic lateral sclerosis (ALS) that are considered monogenic, associated with mutations in SOD1, TARDBP, or FUS [18]”.
However, we appreciate the referee’s suggestion and have revised the text to discuss neurodegenerative diseases with protein aggregation in terms of their familial and sporadic forms. We now highlight that these diseases may have both origins, and that familial cases are linked to genetic mutations, while sporadic cases are influenced by a combination of environmental factors, genetic susceptibility, and age-related changes. This revision ensures greater clarity and specificity, addressing the referee’s concern about the vagueness of the original discussion.
Revised version:
“NDDs characterized by protein aggregation can manifest in both familial and sporadic forms. Familial cases are often linked to specific genetic mutations that predispose individuals to protein aggregation, while sporadic forms arise from a combination of genetic susceptibility, environmental factors, and age-related changes in protein homeostasis. In the hereditary forms of this diseases, mutations in genes, including single nucleotide polymorphisms (SNPs), alternative splicing variants, and repeat expansions, can result in the production of proteins that are prone to aggregation, which in turn enhances their collapse within aggregates.”
Also, we have revised this sentence [lines 101-102] to clarify that mutations in genes lead to the production of aggregation-prone proteins, rather than implying that the genes inherently encode such proteins. This revision ensures the intended meaning is accurately conveyed
Previous text:
“Mutations in genes encoding prone-to-aggregate proteins can enhance protein collapse within aggregates
Revised version:
“[…] mutations in genes, including single nucleotide polymorphisms (SNPs), alternative splicing variants, and repeat expansions, can result in the production of proteins that are prone to aggregation, which in turn enhances their collapse within aggregates”.

Reviewer 3 Report
The usefulness of the model of neurodegenerative diseases using C. elegans has been demonstrated, and although I thought it was a niche, I read it with interest. I think it is well written, but there may be a lack of impartiality in some places because of the overemphasis on the advantages of C. elegans. In addition, I feel that there are many citations of review. For specific phenomena, please cite original research papers as much as possible.
The points that caught my attention are as follows.
The sentences in lines 61 to 62 are redundant.
The protein names should be listed their official names, too.
I probably understand the arguments in lines 102-107, but since there are also mutations related to aggregate formation in AD and PD, doesn't this sentence create a misunderstanding? Also, it is impossible to deny the possibility that several environmental factors are involved in ALS, etc. Notably, most cases of ALS are sporadic. Many factors may affect the disease onset. I feel that the things the authors want to write about in this part are vague. I recommend that several diseases be discussed separately into sporadic and familial forms.
Although Aβ forms oligomers within cells, aggregates are formed outside the cell; therefore, the sentence in Line 137-9 is not accurate.
I think Figure 1 is well written, but I thought the font was small in some parts and difficult to read.
Occasionally, the authors’ enthusiasm for emphasizing the usefulness of C. elegans may exaggerate their expression. For example, the expression 'closely mirroring the pathological hallmarks of AD' in the text from Line 304 is an exaggeration. The nematode model authors cited is artificial in that it expresses Aβ intracellularly, and it mainly looks at intracellular pathology, so I do not think it is close to human pathology. It can be said looking at a fairly limited pathology as a model. The authors also write that AD is a multifactorial disease. If the authors can clearly distinguish between what can and cannot be done using the nematode model and explain it, they will be able to show that they are writing from a fair and objective standpoint in their review article, which will enhance its credibility.
Line 331 describes the conservation of the lipid metabolism mechanism in nematodes, but the reviewer is somewhat skeptical and believes that the conserved lipid metabolism pathway may be limited. Can you provide more specific evidence on how much is conserved and how it is conserved?
Please check whether citation (57) in Line 240 is what the author intended. In addition, it is inappropriate to cite a review when writing about a specific phenomenon. I also feel that this review tends to cite many reviews, so it would be better to be careful about other parts of the paper, as well to see if you can cite a specific paper.
Can Line381 be quoted? Quotation 78 seems to be a work on nematodes, but if the assertion in line 381 is included, it is fine as it is.
I think Table 1 is a good attempt, but in what order are they in? I thought it was a difference in the route of the disorder, but it was not categorized very well, so I could not understand the author's intentions. Why does it not make it easier to understand by arranging them by disease, etc??
Again, I think there are some limitations to the nematode model, such as whether it can reflect the aging process of humans over 80 years old. For example, it would be difficult to reproduce the pathological conditions of Aβ plaques, which form over a period of more than 10 years. Some of these limitations do not go against the authors' desire to demonstrate the usefulness of the nematode model, and I think it shows the authors' impartiality. It would be better to list some of these limitations.
Author Response
Referee 3#
Major comments
This is a timely and informative review of lipid metabolism in C. elegans models of neurodegenerative diseases. It is an important addition to the literature. The focus is mostly on lipid peroxidation, with some info on beta-oxidation. I wonder if having a figure differentiating the two would be informative. Another question is whether lipid oxidation genes themselves are linked to neurodegenerative disorders, directly or as risk factors.
We sincerely thank the referee for their positive comments and this thoughtful suggestion. To address their comment, we have added a new panel to figure 1 that visually differentiates beta-oxidation from lipid peroxidation. This figure is very schematic but highlights the differences between the pathways and processes involved, providing readers with a clearer understanding of these mechanisms and their relevance to neurodegeneration. We believe this addition significantly enhances the clarity and comprehensiveness of the manuscript
Lines 344 - 393:
Thank you for your insightful suggestion regarding the inclusion of information on gene variants involved in lipid oxidation and their association with neurodegeneration or related processes. In response, we have incorporated additional details on this topic into the manuscript. This new section highlights gene variants and their roles in lipid oxidation pathways, emphasizing their relevance to neurodegenerative processes as suggested.
Revised text:
1.4. Key Genes Linking Lipid Dysregulation to Neurodegenerative Disorders.
Certain genes associated with lipid metabolism, particularly lipid oxidation, have been implicated as risk factors or contributors to neurodegeneration. Variants of these genes, as well as isoforms of the proteins they encode, have been identified in patients and experimental models exhibiting neurodegeneration alongside disturbances in lipid metabolism, including elevated lipid oxidation and peroxidation, as highlighted below.
For example, a well-known variant of the apolipoprotein E gene (APOE), apoE4, has been described to pose a strong risk factor for AD ((Chapter 22 - Genetics of neurodegenerative diseases: an overview) https://doi.org/10.1126/science.8346443; https://doi.org/10.1073/pnas.90.5.1977). The APOE protein is implicated in lipid and fatty acid metabolism, and this variant has been associated to an impairment of this processes (https://doi.org/10.1016/j.celrep.2020.108572 ). Notably, apoE4 has also been shown to increase nitric oxide release in humanized APOE mice and human microglia (https://doi.org/10.1016/s0197-4580(02)00016-7), while metabolic shifts towards lipid oxidation have been observed in humanized mice expressing apoE4 (https://doi.org/10.1038/ijo.2016.93).
Another example is the PARK7 gene, encoding the protein DJ-1. DJ-1 has been associated to the development of PD and participates in oxidative stress protection (https://doi.org/10.1111/cge.12841; https://doi.org/10.3390/cells13040296). Likewise, a mutation in the SCP2 gene, encoding the SCPx enzyme, was present in a patient who exhibited progressive neurodegeneration, cardiac dysrhythmia, and metabolic abnormalities, including altered fatty acid levels and disrupted β-oxidation pathways. Pharmacological treatments like fenofibrate and 4-hydroxytamoxifen increased SCPx levels and improved certain metabolic markers, suggesting potential therapeutic strategies for SCPx deficiency (DOI: 10.1186/s40246-022-00408-w). Similarly, LRRK2, has also been associated to PD. LRRK2, which encodes the Leucine-rich repeat kinase 2, is involved in lipid metabolism as it regulates the carnitine palmitoyltransferase 1A (CPT1A), the critical enzyme of β-oxidation.
Concerning enzymes involved in the peroxidation process, changes in their sequence or expression levels have been associated with neurodegeneration. Take, for example, cyclooxygenase (COX), an enzyme of the peroxidation pathway, whose enhanced expression has been associated with several neurodegenerative diseases, such as PD (10.1006/mcne.2000.0914; 10.1016/s0006-8993(02)04174-4; 10.1073/pnas.0837397100) , AD (DOI: 10.1093/jnen/61.8.678; DOI: 10.1016/s0197-4580(01)00303-7); and ALS [DOI: 10.1212/wnl.57.6.952; PMID: 11079544; DOI: 10.1212/wnl.58.8.1277]. Similarly, the mammalian reticulocyte 15-LOX-1 is the major enzyme which is responsible for membrane lipid peroxidation and metabolites apoptosis inducers [https://doi.org/10.1093/jnci/92.14.1136]. This enzyme is widely expressed in the CNS [http://dx.doi.org/10.1038/cddis.2013.86] and increased activity of 15-LOX-1 has been in aged brains during inflammation and neurodegenerative diseases such as AD, and PD [http:// dx.doi.org/10.1111/j.1471-4159.2007.04742.x., http://dx.doi.org/10.1038/mp.2014.170].
Detail comments
- I don't think there is a complete representation of the worm models in Figure 2B, as outlined in Table 1.
We thank the referee for their comment. The aim of Figure 2B was to highlight some of the most used C. elegans models of NDDs; however, we acknowledge that there are many additional models, including those listed in Table 1 and others not included. Our intention was not to create an exhaustive figure but rather to provide a representative selection.
- Starting at line 351, the section about Drosophila glia seems out of place. Why is Drosophila being discussed here and not worms? If it is because there is limited info about C. elegans glia, then why are flies being discussed and not mammals?
We sincerely thank both referees for their valuable feedback. Our intention with this review was to highlight C. elegans as a valuable tool for exploring the mechanisms underlying lipid peroxidation and neurodegeneration, specially for future studies on the field. However, upon reviewing the manuscript, we realized that including Drosophila studies in the context of LDs in neurons may have caused some confusion, as the focus of our review is on the findings in C. elegans. To address this, we have restructured the section to first present the findings from C. elegans regarding lipid droplet formation and its protective effects in neurons, followed by a brief mention of similar findings in Drosophila and other models like cultured hippocampal neurons. We initially included the studies carried on other models as they were relevant to the broader discussion on lipid droplet dynamics, but we now see that separating these findings more clearly in the context of different models will help avoid confusion. We hope these revisions better clarify our focus on C. elegans while still acknowledging the valuable contributions from other models.
Previous text:
The origin and role of glial LDs have been explored more recently. In Drosophila, glial LDs act as a niche for neuroblasts and protect against oxidative stress, particularly under hypoxic conditions. The incorporation of polyunsaturated fatty acids (PUFAs) into neutral lipids stored in LDs helps reduce ROS damage by preventing the toxic peroxidation of these fatty acids. This process mitigates oxidative stress in neuroblasts, which would otherwise be exacerbated by PUFA peroxidation [66,67]. Additionally, when Drosophila neurons face ROS insult or mitochondrial dysfunction, they increase lipid production through SREBP-mediated lipogenesis. However, instead of forming LDs, lipids are transferred to neighboring glia to form LDs. In cultured hippocampal neurons, excess fatty acids are transferred to astrocytes via ApoE-associated lipid particles and stored in astrocyte LDs, where they are β-oxidized in mitochondria. This β-oxidation helps protect neurons during periods of enhanced activity by reducing oxidative stress and maintaining cellular energy balance [68]. These findings suggest that glial LDs play a protective role for neurons by reducing oxidative stress, although it remains unclear why neurons do not autonomously form LDs under stress conditions.
Revised version:
The role of glial LDs in this context has been explored more recently. Mutations in key lipolysis genes in C. elegans have been shown to lead to the appearance of lipid droplets (LDs) in neurons, which offer protection from hyperactivation-triggered neurodegeneration. This protection is associated with a mild reduction in touch sensation. Additionally, reduced biosynthesis of polyunsaturated fatty acids (PUFAs) and impaired lipolysis work synergistically to enhance PUFA partitioning into triacylglycerol rather than phospholipids, providing further neuronal protection. These findings highlight the critical role of neuronal lipolysis in regulating neuronal function and protecting against neurodegeneration [doi: 10.15252/embr.202050214]
Similarly, in Drosophila, glial LDs have been shown to protect neuroblasts from oxidative stress, particularly under hypoxic conditions. The incorporation of PUFAs into neutral lipids stored in LDs helps reduce ROS damage by preventing PUFA peroxidation. In response to ROS insult or mitochondrial dysfunction, Drosophila neurons increase lipid production via SREBP-mediated lipogenesis, with lipids being transferred to neighboring glia to form LDs [66,67]. Furthermore, in cultured hippocampal neurons, excess fatty acids are transferred to astrocytes via ApoE--associated lipid particles, where they are stored in astrocyte LDs and β-oxidized in mitochondria. This β-oxidation helps protect neurons from oxidative stress and maintain energy balance during periods of enhanced neuronal activity[68]. These findings suggest that glial LDs play a protective role for neurons by reducing oxidative stress, although it remains unclear why neurons do not autonomously form LDs under stress conditions
Table 1 should have a title for the first column, and the other titles should have the first letter capitalized.
We appreciate the comments, and following the recommendations we have modified the table, giving a logical order to the publications by pathology and we have additionally added a couple more references that we consider relevant (also included in the text).
The text is small for lines 457-467.
We have addressed this as suggested by the referee.
Some grammar issues:
- line 46-7. Possessives for the disease names are missing.
- line 65. The abbreviation for C. elegans should be done earlier.
- line 104. Nomenclature for human genes should be consistent (italics for gene, etc.)
- line 111. Grammar and syntax issues.
- line 278. consistency for beta versus greek symbol for beta
- lines 505, 509 Grammar and syntax issues.
We thank the referee for pointing out these mistakes. We have corrected
OTHER CHANGES:
- Lines 81-87: We revised this section as the original phrasing was too vague. The updated text better reflects our intended message.
Previous text:
“we discuss the molecular mechanisms underlying lipid oxidation-mediated neurotoxicity and protein misfolding taking advantage of C. elegans models of neurodegeneration. Understanding this intricate relationship offers insights into novel therapeutic targets and potential interventions for mitigating oxidative stress and restoring cellular homeostasis in NDD. Targeting lipid metabolism pathways may provide innovative strategies to alleviate neuronal dysfunction and slow disease progression, holding promise for improving outcomes in patients with NDD.”
Revised version:
“we aim to explore the dual role of lipid oxidation in neurotoxicity within these diseases , both by exacerbating protein misfolding and by damaging lipids, which disrupts their structural integrity and impairs their functions. Additionally, we highlight the potential of C. elegans models as a simple yet robust platform for further investigating this phenomenon. Understanding this intricate relationship offers insights into targeting specifically lipid oxidation for alleviating neuronal dysfunction and slow disease progression, holding promise for improving outcomes in patients with NDD.”
- Section 2:. Based on the referees’ suggestions, we have divided the section into two distinct subsections. The first subsection focuses on the advantages and challenges of using elegans as a model for studies on the role of lipid oxidation in Neurodegeneration, while the second highlights the specific bibliography that explore the role of lipid peroxidation in neurodegenerative diseases using C. elegans. This division allows us to provide a clearer and more organized discussion, ensuring that both the broader context and specific findings are addressed in a way that better serves the purpose of this review. Also, we have changed Figure 2’s disposition on the manuscript to better fit and illustrate our point from the discussion to the section number 2.1 “Advantages and Challenges of Using C. elegans for Lipid Peroxidation Studies.”, in which the topic of the figure is discussed.
- Lines 250-268: While the referees did not specifically request changes to this section, we felt the original statement about lipid peroxides and protein misfolding incomplete. To improve clarity, we expanded on how lipid peroxides promote protein destabilization and misfolding, enhancing the manuscript's depth and precision.
Previous text:
“Lipid peroxides can exacerbate protein misfolding by promoting oxidative modifications of proteins, which can alter their structure and function. In turn, these misfolded proteins can further disrupt cellular homeostasis by interfering with processes involved in maintaining proteostasis, such as the UPS and autophagy”.
Revised version:
Lipid peroxides can exacerbate protein misfolding by promoting oxidative modifications of proteins, which can alter their structure and function. When proteins misfold, hydrophobic regions of cysteine residues can be exposed on the protein surface, making them vulnerable to oxidation by reactive oxygen species (ROS) and other oxidants [https://doi.org/10.1146/annurev.bi.62.070193.004053]. Oxidation of cysteine residues can result in the formation of disulfide bonds and mixed disulfide bonds. Additionally, oxidation can disrupt non-covalent interactions within proteins, cause peptide chain fragmentation, promote protein cross-linking, and oxidize specific side chains. These changes collectively contribute to protein destabilization and further misfolding [PMID: 4034546; https://doi.org/10.1016/S0021-9258(18)48018-0]. In turn, these misfolded proteins can further disrupt cellular homeostasis by interfering with processes involved in maintaining proteostasis, such as the UPS and autophagy
- Discussion and conclusions sections:
We sincerely appreciate the thoughtful and constructive feedback provided by the reviewers, which has greatly enhanced the clarity and depth of our manuscript. Based on these valuable suggestions, we have revised both the discussion and conclusion sections to better align with the new approach of the review. We hope that these revisions adequately address the reviewers’ insightful suggestions and strengthen the manuscript's overall impact. Thank you once again for your valuable feedback and for the opportunity to improve our work.
Lines 595-624:
Revised version:
“Lipid peroxidation has emerged as a critical factor in the pathogenesis of neurodegenerative disorders, serving as a convergence point for oxidative stress and cellular damage. Processes such as ferroptosis, a regulated form of cell death driven by iron-dependent lipid peroxidation, highlight the vulnerability of neurons to oxidative insults [67]. In the context of NDDs, the accumulation of lipid peroxides contributes to the disruption of membrane integrity, reducing fluidity and impairing cellular signaling, both of which are essential for neuronal function and survival [77]. Additionally, it promotes protein misfolding [161], which can lead to a feedback loop that exacerbates cellular damage.
Mutations in genes associated with lipid metabolism, antioxidant defenses, and iron homeostasis have been linked to heightened oxidative stress and accelerated neuronal damage. For instance, the enhanced expression of enzymes involved in lipid peroxidation pathways, such as cyclooxygenases, has been identified in models of neurodegenerative diseases [97.98]. Moreover, the use of lipids as biomarkers for NDDs progression has already been suggested [160], nonetheless, the heterogeneity of neurodegenerative disorders necessitates a personalized approach to therapeutic development, as the contribution of lipid peroxidation may vary across different diseases and patient populations.
These findings suggest that dysregulated lipid peroxidation may not only be a hallmark of neuronal damage but also act as a driving force for disease progression. Intervention strategies targeting lipid peroxidation have shown promise in mitigating neurodegenerative processes. ROS-scavenger compounds, including ferrostatins have demonstrated efficacy in reducing ferroptosis and preserving neuronal integrity [127]. These compounds function by neutralizing lipid radicals and preventing the propagation of peroxidative damage, offering a potential therapeutic avenue for halting or slowing disease progression. Nevertheless, challenges remain in translating these findings into clinical applications. The complex interplay between lipid peroxidation and other pathological processes, such as protein aggregation and mitochondrial dysfunction, warrants further investigation.
In this regard, [...]”
Lines 676–695:
Previous text:
“In summary, targeting lipids and the products of the oxidized lipids by free radicals in the treatment of neurodegenerative diseases not only addresses lipid dysregulation but also positively impacts proteostasis. By reducing oxidative stress, enhancing protein folding, boosting degradation pathways, and promoting cell survival, antioxidant lipid-based therapies hold promise for mitigating the detrimental effects of protein misfolding and aggregation, ultimately improving neuronal health and function. Furthermore, strategies that modulate lipid metabolism to reduce the accumulation of toxic lipid peroxides may enhance the efficacy of existing proteostasis pathways, offering a multifaceted approach to slowing or potentially reversing neurodegeneration.
Understanding lipid metabolism in neurodegenerative diseases is opening new avenues for therapeutic interventions.”
Revised version:
“Lipid peroxidation plays a central role in the pathogenesis of NDDs, acting as a key driver of oxidative stress and cellular damage. Its contributions to processes like ferroptosis, protein misfolding, and disrupted neuronal signaling underscore its critical importance as both a marker and a therapeutic target.
As demonstrated by this review, the ease of generating mutant C. elegans strains for nearly any gene of interest establishes this model as a powerful platform for investigating phenotypes and their underlying pathways. Furthermore, the capacity to integrate these genetic models with pharmacological treatments and dietary supplementation assays further enhances the versatility of C. elegans as a research tool. This combined approach facilitates the systematic exploration of gene-environment interactions and the evaluation of potential therapeutic interventions, providing valuable insights into disease mechanisms and treatment strategies.
By leveraging this model, the field can make significant progress in understanding the interplay between lipid peroxidation and neurodegeneration. Future research should focus on integrating advanced methodologies, such as OMICs approaches and high-resolution imaging performed in patients, to further elucidate the role of lipid peroxidation and its downstream effects. These efforts have the potential to identify novel therapeutic targets and develop personalized interventions that mitigate lipid peroxidation, ultimately slowing the progression of neurodegenerative disorders.”

Round 2
Reviewer 2 Report
The authors have answered my concerns and have provided an interesting review article that should be relevant to a wide audience.
The inclusion of the oxidation figure was helpful.
Author Response
We thank very much the referee fr his/her kind comments
Reviewer 3 Report
While I admit that most of the revisions were made in good faith, my points in several areas were deleted and unanswered.
I do not see this as a problem, as each researcher may have his/her own ideas.
However, I think 'closely mirroring' in Lines 415-416 is a bit of a quip, so please change it to something more moderate. As mentioned, C. elegans lacks some of the Aβ-producing enzymes in humans and cannot be an accurate aggregation model.
I think it is better to show the limitations of the nematode model to some extent.
However, I think 'closely mirroring' in Lines 415-416 is a bit of a quip, so please change it to something more moderate. As mentioned, C. elegans lacks some of the Aβ-producing enzymes in humans and cannot be an accurate aggregation model.
I think it is better to show the limitations of the nematode model to some extent.
Author Response
Thank you, again, to the referees and editor.
We are sorry about this mistake. We thought we did respond to every single point of the three referees.
While I admit that most of the revisions were made in good faith, my points in several areas were deleted and unanswered.
I do not see this as a problem, as each researcher may have his/her own ideas.
We thank the reviewer for the generosity in his appreciation of the revision.
However, I think 'closely mirroring' in Lines 415-416 is a bit of a quip, so please change it to something more moderate. As mentioned, C. elegans lacks some of the Aβ-producing enzymes in humans and cannot be an accurate aggregation model.
We agree with the referee. This expression was not the best choice to compare models to phenotypic traits of the disease.
We have changed 'closely mirroring' for “presenting somehow similar traits to”
I think it is better to show the limitations of the nematode model to some extent.
I am not sure the referee got our rebuttal letter in which we reply to all three referees. In the letter we explain that we have added plenty of limitations of the C. elegans models:
This is what we wrote in our original rebuttal letter:
Lines 348-360: We sincerely appreciate the referee’s valuable suggestions. In response, we have expanded the section discussing the limitations of the model organism, such as the challenges in isolating specific cells or tissues for biochemical analyses, the absence of certain vertebrate tissues and organs, the potential for artifactual neurodegeneration models, and the gene redundancy issue that complicates knockout generation. In addition, we have stated its short life cycle, which enables rapid phenotype progression and the generation of consecutive generations within a few weeks. We believe these additions provide a more balanced and comprehensive discussion of the strengths and limitations of C. elegans in studying neurodegenerative diseases.
Text added:
However, there are limitations to consider when using C. elegans as a model. Isolating specific cells or tissues for biochemical analyses is challenging, though not impossible, as techniques such as single-worm and single-cell approaches have been developed. Moreover, C. elegans lacks some tissues and organs found in vertebrates, such as eyes and a heart—though the pharynx is considered an analogous structure—and a liver, which is functionally substituted by its intestine. Some neurodegeneration models in C. elegans are also considered artefactual, as the toxic gene products being studied do not naturally exist in these organisms. In addition, gene redundancy poses challenges; some mammal and human genes have many orthologs in C. elegans with non-identical functions, making it harder to generate knockouts. While the homology between C. elegans and higher organisms is high, it is not identical, which can limit direct comparisons in certain contexts."
We hope that now the paper is suited for Antioxidants
Thank you, again, to the referees and editor.
We are sorry about this mistake. We thought we did respond to every single point of the three referees.
While I admit that most of the revisions were made in good faith, my points in several areas were deleted and unanswered.
I do not see this as a problem, as each researcher may have his/her own ideas.
We thank the reviewer for the generosity in his appreciation of the revision.
However, I think 'closely mirroring' in Lines 415-416 is a bit of a quip, so please change it to something more moderate. As mentioned, C. elegans lacks some of the Aβ-producing enzymes in humans and cannot be an accurate aggregation model.
We agree with the referee. This expression was not the best choice to compare models to phenotypic traits of the disease.
We have changed 'closely mirroring' for “presenting somehow similar traits to”
I think it is better to show the limitations of the nematode model to some extent.
I am not sure the referee got our rebuttal letter in which we reply to all three referees. In the letter we explain that we have added plenty of limitations of the C. elegans models:
This is what we wrote in our original rebuttal letter:
Lines 348-360: We sincerely appreciate the referee’s valuable suggestions. In response, we have expanded the section discussing the limitations of the model organism, such as the challenges in isolating specific cells or tissues for biochemical analyses, the absence of certain vertebrate tissues and organs, the potential for artifactual neurodegeneration models, and the gene redundancy issue that complicates knockout generation. In addition, we have stated its short life cycle, which enables rapid phenotype progression and the generation of consecutive generations within a few weeks. We believe these additions provide a more balanced and comprehensive discussion of the strengths and limitations of C. elegans in studying neurodegenerative diseases.
Text added:
However, there are limitations to consider when using C. elegans as a model. Isolating specific cells or tissues for biochemical analyses is challenging, though not impossible, as techniques such as single-worm and single-cell approaches have been developed. Moreover, C. elegans lacks some tissues and organs found in vertebrates, such as eyes and a heart—though the pharynx is considered an analogous structure—and a liver, which is functionally substituted by its intestine. Some neurodegeneration models in C. elegans are also considered artefactual, as the toxic gene products being studied do not naturally exist in these organisms. In addition, gene redundancy poses challenges; some mammal and human genes have many orthologs in C. elegans with non-identical functions, making it harder to generate knockouts. While the homology between C. elegans and higher organisms is high, it is not identical, which can limit direct comparisons in certain contexts."
We hope that now the paper is suited for Antioxidants

Round 3
Reviewer 3 Report
I have no additional comments.
There are no further corrections to be made for my part.
Author Response
Thank you very much for the kind comments of the referee